# Assessment of 3D-Printed Polycaprolactone, Hydroxyapatite Nanoparticles and Diacrylate Poly(ethylene glycol) Scaffolds for Bone Regeneration

**DOI:** 10.3390/pharmaceutics14122643

**Published:** 2022-11-29

**Authors:** Ana Catarina Sousa, Sara Biscaia, Rui Alvites, Mariana Branquinho, Bruna Lopes, Patrícia Sousa, Joana Valente, Margarida Franco, José Domingos Santos, Carla Mendonça, Luís Atayde, Nuno Alves, Ana Colette Maurício

**Affiliations:** 1Veterinary Clinics Department, Abel Salazar Biomedical Sciences Institute (ICBAS), 4050-313 Porto, Portugal; 2Animal Science Studies Centre (CECA), Agroenvironment, Technologies and Sciences Institute (ICETA), University of Porto (UP), Rua D. Manuel II, Apartado 55142, 4051-401 Porto, Portugal; 3Associate Laboratory for Animal and Veterinary Sciences (AL4AnimalS), Faculdade de Medicina Veterinária (FMV), Universidade de Lisboa, Avenida da Universidade Técnica, 1300-477 Lisbon, Portugal; 4Centre for Rapid and Sustainable Product Development (CDRSP), Polytechnic of Leiria, 2411-901 Leiria, Portugal; 5REQUIMTE-LAQV, Departamento de Engenharia Metalúrgica e Materiais, Faculdade de Engenharia, Universidade do Porto, Rua Dr. Roberto Frias, 4200-465 Porto, Portugal

**Keywords:** bone regeneration, critical bone defects, hydroxyapatite nanoparticles, polycaprolactone, diacrylate poly(ethylene glycol), scaffolds

## Abstract

Notwithstanding the advances achieved in the last decades in the field of synthetic bone substitutes, the development of biodegradable 3D-printed scaffolds with ideal mechanical and biological properties remains an unattained challenge. In the present work, a new approach to produce synthetic bone grafts that mimic complex bone structure is explored. For the first time, three scaffolds of various composition, namely polycaprolactone (PCL), PCL/hydroxyapatite nanoparticles (HANp) and PCL/HANp/diacrylate poly(ethylene glycol) (PEGDA), were manufactured by extrusion. Following the production and characterisation of the scaffolds, an in vitro evaluation was carried out using human dental pulp stem/stromal cells (hDPSCs). Through the findings, it was possible to conclude that, in all groups, the scaffolds were successfully produced presenting networks of interconnected channels, adequate porosity for migration and proliferation of osteoblasts (approximately 50%). Furthermore, according to the in vitro analysis, all groups were considered non-cytotoxic in contact with the cells. Nevertheless, the group with PEGDA revealed hydrophilic properties (15.15° ± 4.06) and adequate mechanical performance (10.41 MPa ± 0.934) and demonstrated significantly higher cell viability than the other groups analysed. The scaffolds with PEGDA suggested an increase in cell adhesion and proliferation, thus are more appropriate for bone regeneration. To conclude, findings in this study demonstrated that PCL, HANp and PEGDA scaffolds may have promising effects on bone regeneration and might open new insights for 3D tissue substitutes.

## 1. Introduction

The population aging leads to a remarkable increase in the number of degenerative diseases, osteogenic disorders, fractures and bone infections [1,2]. Although bone tissue can heal itself to a certain extent following bone pathology, when it concerns critical-sized defects (size above about 3 cm), it may not be fully restored [3,4,5]. Particularly in such cases, the reconstruction of bone defects, with mechanical integrity to the original surrounding bone tissues, is essential for a patient’s rehabilitation [6]. Therefore, the search for new solutions has focused on tissue engineering through the development of three-dimensional (3D) structures, namely scaffolds for the regeneration of bone tissues [7,8,9]. Some properties must be considered when producing scaffolds for bone regeneration: (a) biocompatibility devoid of unchained negative biological response in the body; (b) osteoconduction to promote cell adhesion and bone growth; (c) biodegradability to ensure controlled replacement of the biomaterial by the neoformed bone; (d) mechanical properties to ensure support during bone bridging; (e) sterility of the material; and (f) appropriate design in terms of porosity, interconnectivity and pore size to provide the cell proliferation and angiogenesis [10,11,12]. In recent years, nanoparticles (e.g., hydroxyapatite, gold and carbon) have been the subject of research as they control the structure of the material at the nanoscale. The materials show a higher resolution of the nanocomposite structure the smaller the size of these nanoparticles [13,14]. In addition to the development of biomaterial support, cells and growth factors have an important role in the formation of biological substitute, so it is necessary to resort to regenerative medicine and tissue engineering [15]. The use of biomaterials with mesenchymal stem cells (MSCs) allows the proliferative and differentiation capacities of the latter to work in synergy with scaffolding properties [1,16].

Today, several additive manufacturing techniques are used to produce complex bone implants, namely, selective laser sintering, selective laser melting, stereolithography, electron beam melting, electrospinning and fused deposition modelling [17]. Fused deposition modelling, commonly known as an extrusion-based process, is a promising 3D-printing and -manufacturing technique in the production of interconnected porous scaffolds [18]. This technique is easy to operate, is safe, reliable and controllable, and the produced structures normally have good mechanical properties [4].

According to the literature, synthetic rigid porous scaffolds have usually been made based on hydroxyapatite nanoparticles (HANp), biphasic calcium phosphate (BCP), beta-tricalcium phosphate (β-TCP) and polycaprolactone (PCL) [19,20,21,22]. PCL has been widely used because of its good biocompatibility, biodegradability, ease of processing (melting point between 55 and 60 °C) and the fact that its blends well with other materials such as ceramics [23,24,25]. Furthermore, HANp, as a biomaterial, presents good stability, biocompatibility and degradability, promotes the adhesion/proliferation of osteoblasts and has the potential to form chemical bonds with the bone itself [26,27,28]. Several studies combined PCL and HANp in scaffolds because of their properties and achieved good results inherent to bone regeneration both in vitro and in vivo [29,30,31,32,33,34,35,36,37,38]. In work by Song and colleagues, the results indicated that cell activities in PCL-HANp scaffolds are higher than in PCL/hydroxyapatite, possibly because of the higher hydrophilicity and porosity of the PCL-N/HA scaffold, than in PCL/hydroxyapatite [29]. Chuenjitkuntaworn and colleagues demonstrated that PCL/HANp scaffolds exhibit higher levels of calcium deposition compared to PCL alone, and that they support the growth of various mesenchymal stem cell types [37]. Furthermore, El-Habashy et al. evaluated biopolymer-based hydrogel scaffolds enhanced with bioactive hybrid hydroxyapatite/polycaprolactone nanoparticles in rabbit tibial bone defects. The results demonstrated that the produced scaffolds supported bone regeneration in vivo, providing adequate biodegradation, biocompatibility and osteogenic/osteoconductive properties [38]. Previous studies also consider diacrylate poly(ethylene glycol) (PEGDA) hydrogel as an effective biomaterial in bone regeneration because of its properties such as strength, photo-crosslinkability, gelation processes, hydrophilicity and cell adhesion [39,40,41,42,43,44]. Kotturi and colleagues assessed the physical, mechanical and biological properties of the PEDGA-PCL scaffold toward tissue engineering applications. The results demonstrated efficacy in the combination of PCL and PEGDA, showing that these scaffolds absorb nutrients over time and can provide an optimal environment for cell survival, adhesion, proliferation and migration [42]. In the study by Liu et al., a mineralised PEGDA/HA hydrogel loaded with Exendin4 (a stable analogue of the gut hormone GLP-1) was produced for the healing process of bone defects, demonstrating good biocompatibility and mechanical properties [39].

For the present study, combined scaffolds of PCL, HANp and PEGDA were produced by an extrusion additive manufacturing system. To the best of our knowledge, this is the first study to incorporate these three materials into a scaffold for bone application. To fill the gap in the literature, the production of these compounds from the incorporation of HANp into synthetic polymers (PCL) with the coating of a photo-crosslinkable hydrogel combines the advantages inherited by each of these components. Therefore, it is expected that, while PCL/HANp provides mechanical support, osteoconductivity and interconnectivity between the pores for cell proliferation, the PEGDA coating provides more hydrophilicity to the structure and better characteristics for cell adhesion. The surface chemistry of 3D-printed scaffolds was characterised using Fourier transform infrared spectroscopy (FTIR) and contact angle measurement. The morphological properties were evaluated by X-ray micro-computed tomography (Micro-CT), scanning electron microscopy (SEM) and energy-dispersive X-ray spectroscopy (EDX). Moreover, compression tests were performed to assess the mechanical response of the scaffolds. Scaffolds were further characterised in vitro, assessing their cytocompatibility properties. Thus, this study aims to lay the groundwork for future research into the use of these three materials (PCL, HANp and PEGDA) for more accelerated and effective bone regeneration.

## 2. Materials and Methods

### 2.1. Materials

PCL (CAPA^®^ 6500) from Perstorp Caprolactones (Cheshire, UK) (Mw: 50 kDa) and HANp (particle size < 200 nm) from Sigma-Aldrich (Saint Louis, MO, USA) were used. Formulations were produced using N,N-dimethylformamide (DMF) from CHEM-LAB (Belgium) and by the solvent casting technique. For the hydrogel formulation, PEGDA, from Sigma-Aldrich (Saint Louis, MO, USA), number average molecular weight (Mn) = 750 (mol) and HEPES solution (Gibco, 15140122) were used. Photopolymerisation was induced using 0.1% *w*/*v* 2-Hydroxy-4′-(2-hydroxyethoxy)-2-methylpropiophenone 98% (Irgacure 2959, Sigma-Aldrich) photo-initiator at UV light (365 nm, 690 mV) exposure.

### 2.2. Sample Design

To develop the scaffolds, protocols were designed relying on the available revised literature [45,46,47,48]. In the present work, a biomanufacturing system (Biomate Project from ANI) developed by the CDRSP-IPLeiria was used [45,48,49,50,51,52,53]. This equipment integrates three biomanufacturing techniques: a micro-extrusion system, a multi-head dispensing system and electrospinning [46]. Three different matrices were produced in this system: (i) PCL scaffolds; (ii) PCL scaffolds with the addition of HANp; and (iii) PCL/HANp scaffolds submerged in PEGDA solution.

For the production of the scaffolds, a nozzle with a diameter of 400 µm was used. The parameters employed were 240 mm/min of deposition velocity, 9 rpm of screw rotation velocity and 85 °C of liquefier temperature. The methodologies used in the production of the scaffolds (diameter: 10 mm, height: 3 mm, and pore size: 380 µm) were as follows:(i)PCL scaffolds:

The PCL was dissolved in DMF (2 mL of DMF for each 0.5 g of PCL) at 80 °C and dried in a controlled environment for 96 h. The membranes were cut into pieces to be subsequently placed in the bioextrusion equipment deposit.

(ii)PCL scaffolds with the addition of hydroxyapatite nanoparticles (HANp):

PCL (60 wt%) and HANp (40 wt%) were dissolved in DMF and dried in a controlled environment. The completely dried membranes were cut into pieces to be subsequently placed in the bioextrusion equipment deposit.

(iii)PCL/HANp scaffolds submerged in PEGDA solution:

For PCL/HANp scaffolds, the same procedure previously mentioned in point ii was employed. The PEGDA (6 wt% in deionised water), Hepes and Irgacure 2959 solution was made by melt blending at 50 °C, using a heating plate with stirrer. Afterwards, the PCL/HANp scaffolds were submerged in the previously made solution. The scaffolds were carefully removed from the solution and were crosslinked using UV light exposure (365 nm) for four minutes. The schematic representation of the procedure is summarised in Figure 1.

### 2.3. Material Characterisation

The physical, chemical and mechanical characterisation of the produced scaffolds was performed: Fourier transform infrared spectroscopy (FTIR), contact angle measurement, micro-computed tomography (Micro-CT), scanning electron microscopy (SEM), energy-dispersive X-ray analysis (EDX), and compression tests.

#### 2.3.1. Fourier Transform Infrared Spectroscopy

To extract qualitative chemical information, samples were analysed using the Bruker Alpha-P ATR FTIR spectrometer (Bruker, Kontich, Belgium) and Opus software. The tests were carried out at room temperature, in a spectral range of 4000–400 cm^−^^1^, with a resolution of 4 cm^−^^1^ in a total of 64 scans.

#### 2.3.2. Contact Angle Measurement

The wettability of the formulations was evaluated by static contact angle measurement at 10 s on a Theta Lite optical tensiometer (Attension, Biolin Scientific, Espoo, Finland). A water droplet was dispensed on the surface of solid samples and the contact angle was measured by OneAttension 2.1 software (Attension).

#### 2.3.3. X-ray Micro-Computed Tomography

Micro-CT scans of the scaffolds were performed using a SkyScan microtomograph model 1174, Brucker (Kontich, Belgium). Scan parameters selection, flat field correction and every operator’s choice during the scan, reconstruction and analysis steps are very important to get the best results. The scan parameters selected for the digitalisations involved in this work were: for PCL only: 50 Kv; 800 µA; 19.6 image pixel size; 4500 ms exposure; averaging frames 3; 0.9 rotation degree; and no filter. For PCLHANp and PCL/HANp/PEGDA: 50 Kv; 800 µA; 19.6 image pixel size; 6500 ms exposure; averaging frames 3; 0.9 rotation degree; and 0.25 Al filter (to increase the photon energy of the beam because of the presence of HANp). The scan duration was 1:10 h. Acquired radiographs were collected on a 1.3 Mp CCD charge-coupled device (CCD) coupled to a scintillator by the lens. The images were then mathematically reconstructed into slices with NRecon v.1.7.0.4 software (Bruker Micro-CT) using a 25% beam-hardening correction, a ring artefact correction of 5 and similar contrast limits among similar specimens. A dataset composed of 573 cross-sections was obtained and a region of interest (ROI) was then selected over a representative amount of 300 slices; the ROI was used for the morphometry analysis made with the help of CTAn v.1.20.3.0 software (Bruker Micro-CT). CTVox v.3.2.0 (Bruker Micro-CT) was the software used to perform a 3D volume rendering of the total dataset, providing a 3D viewing environment and 3D images; parameters for light and opacity control were selected.

#### 2.3.4. Scanning Electron Microscopy and Energy-Dispersive X-ray Analysis

To analyse the filament and pore morphology, scaffolds from each experimental group were analysed by SEM using Vega3 Tescan equipment (Tescan, Brno, Czechia), operating at an accelerating voltage of 20 kV, at variable magnifications and with a working distance of around 10 mm. The samples were fixed on a brass stub using double-sided tape and then made electrically conductive by coating with gold/palladium (Au/Pd) thin film, by sputtering, using the sputter coater equipment for 45 s with 5 cm of distance between the target and the sample and 20 mA (SC7620 Quorum Technologies, Lewes, UK). The samples were also analysed using EDX (Xflash 6|30 from Brucker, Billerica, MA, USA).

#### 2.3.5. Mechanical Analysis

Compression tests were performed to evaluate the mechanical properties of each scaffold. The tests were conducted according to ASTM STP 1173 standards [54], using a TA.XTplusC (Stable Micro Systems, Godalming, UK) with an extension rate of 0.6 mm/min. Mechanical testing was carried out using six scaffold samples, with a diameter of 10 mm and a height of 3 mm. Stress (MPa) data was computed from load-displacement measurements.

### 2.4. In Vitro Tests

#### 2.4.1. Cell Culture and Maintenance

The human dental pulp stem/stromal cells (hDPSCs) used in this study were sourced from AllCells, LLC (Cat. DP0037F, Lot No. DPSC090411-01). These were maintained in MEM α, GlutaMAX™ supplement, nucleoside-free (Gibco, 32561029). This medium was supplemented with 10% (*v*/*v*) foetal bovine serum (FBS) (Gibco, A3160802), 100 IU/mL penicillin, 0.1 mg/mL streptomycin (Gibco, 15140122), 2.05 µm/mL amphotericin B (Gibco, 15290026) and 10 mM HEPES buffer solution (Gibco, 15630122). DPSCs were maintained in standard conditions, namely at 37 °C in 80% humidified atmosphere and 5% CO_2_.

#### 2.4.2. Cytocompatibility Assessment

Prior to the cytocompatibility tests, all scaffolds in this study were sterilised by gamma radiation (25 kGy) in a Red Perspex dosimeter. Then, the samples were tested with hDPSCs using the PrestoBlue^TM^ viability to assess the impact of the scaffolds on cell adhesion and viability. This reagent is a commercially available, ready-to-use, water-soluble, resazurin-based solution (7-hydroxy-3H-phenoxazin-3-one-10-oxide). The active cells reduce this compound into resazurin, a process that is accompanied by a change in the colour and fluorescence of the solution. Consequently, absorbance measurements of the solution indicate viability, thus allowing quantitative measurement of cell proliferation. Due to the reduction in the compound by the viable cells, the solution colour changes from blue to a reddish tone. Thus, changes in cell viability/proliferation were assessed by corresponding changes in absorbance measurements [55].

In brief, scaffolds were seeded using dynamic seeding, i.e., these were incubated in cell suspension (density of 2.5 × 10^5^ per scaffold) on a roller bench all overnight at 37 °C in an 80% humidified atmosphere and a 5% CO_2_ environment. Later, the seeded scaffolds were moved to a 24-well non-adherent plate and submerged in fresh complete medium. Presto Blue^TM^ evaluation was performed at different time points: 24, 72, 120 and 168 h. Thus, for each time point, culture media was removed from each well and replaced with fresh complete medium with 10% (*v*/*v*) of the PrestoBlue^TM^ reagent (Invitrogen, A13262). Cells were incubated for 1 h at 37 °C in an 80% humidified atmosphere and 5% CO_2_. Supernatant media were transferred to a 96-well plate and absorbance was read at 570 and 595 nm. Changes in cell viability were detected by absorbance spectroscopy in a spectrophotometer, Multiskan^TM^ FC Microplate Photometer (Thermo Scientific^TM^, 51119000). For each well, the absorbance at 595 nm (normalisation wavelength) was subtracted from the absorbance at 570 nm (experimental result). The corrected absorbance is obtained by subtracting the mean of the control wells for each experimental well. The cells were then washed in Dulbecco’s phosphate buffered saline (DPBS, Sigma^®^, D8537) to remove any residual PrestoBlue^TM^ and then fresh culture medium is replenished in each well. The data were analysed and subsequently normalised to the mean of the gold standard (PCL group), presented as % viability inhibition.

### 2.5. Statistical Analysis

Statistical analysis was performed by using GraphPad Prism^®^ (version 8.4.0 for Mac OS, La Jolla, CA, USA). The results were presented as mean ± standard error of the mean (SE). Group comparisons were accomplished by one-way ANOVA followed by Tukey’s multiple comparisons test. The differences were only considered statistically significant when *p* ≤ 0.05. Significant results are indicated according to *p* values with one, two, three or four of the symbols (*) representing 0.01 < *p* ≤ 0.05, 0.001 < *p* ≤ 0.01, 0.0001 < *p* ≤ 0.001 and *p* ≤ 0.0001, respectively.

## 3. Results

### 3.1. Scaffolds Production

In this study, three scaffolds with different composition, namely, PCL, PCL/HANp and PCL/HANp/PEGDA, were manufactured by extrusion. The samples were produced by filament deposition with a 10 mm diameter, 3 mm height and 380 µm pore size. Figure 2 shows 3D-printed scaffolds representative of each group.

According to the results (Figure 2a–c), the 3D-printed scaffolds were successfully produced. The filaments, in all groups, seem to be well-coordinated and -positioned.

### 3.2. Material Characterisation

For the assessment of the functional groups, the samples were analysed by FTIR spectroscopy. Therefore, segments from each group (PCL, PCL/HANp, PCL/HANp/PEGDA and PEGDA) were analysed (Figure 3). In the PCL, PCL/HANp and PCL/HANp/PEGDA groups are represented the characteristic absorption bands of PCL, which are asymmetric CH_2_ at 2943 cm^−^^1^, symmetric CH_2_ at 2865 cm^−^^1^, C=O at around 1720 cm^−^^1^, and C-O and C-C stretching corresponding to the crystalline phase at 1239 cm^−^^1^ [49,56,57]. Regarding HANp, in the PCL/HANp and PCL/HANp/PEGDA sample the peaks corresponding to phosphates (ν1 and ν3) are around 1300 cm^−^^1^. Furthermore, the P-O stretching is represented at 1088, 600 and 568 cm^−^^1^ [49,56,58]. Finally, in the PCL/HANp/PEGDA sample, the characteristic peaks of PEGDA are represented at 1638 cm^−1^ and 910 cm^−1^ (the double peak is due to elongation of the vinyl groups) and 1720 cm^−1^. Another characteristic peak of PEGDA is the OH stretching represented between 3500 and 3400 cm^−1^ [59,60].

The wetting tendency of the samples was assessed by measuring the contact angle and is shown in Figure 4.

This characterisation test revealed that the PCL scaffolds (85.10 ± 3.54°) presented a slightly higher contact angle than the PCL/HANp scaffolds (80.48 ± 1.01°). Nevertheless, the PCL/HANp/PEGDA group showed a significantly lower contact angle (15.15 ± 4.06°) than the other groups. Considering the results, the addition of HANp and PEGDA seems to have decreased the contact angle of the samples.

The internal and external morphologies (Figure 5) and the porosity (Table 1) of the scaffolds were studied by using X-ray micro-computed tomography.

All scaffold groups were produced presenting interconnected channel networks and good geometric accuracy (Figure 4). Based on the results of porosity (%) of the scaffolds (Table 1), there is no evidence of significant statistical differences between the different scaffolds.

Concerning SEM analysis, it was used to visualise the filaments and pores of the prepared scaffolds, and details about their morphology and topography, as represented in Figure 6.

According to the results, the scaffolds seem to have been successfully produced, revealing interconnected porosity and well-accurate filaments (Figure 6A.1–C.1). The structural characteristics of the scaffolds were also analysed by measuring the filaments and pore size for each experimental group. No statistical differences were found between the filaments of the three experimental groups. These measured approximately 400 µm in diameter, which corroborates with the conception parameters of the scaffolds. Furthermore, all scaffolds presented interconnected and square pores with diameters of 380 µm (with no statistical differences between the groups). At high magnifications, the PCL scaffolds revealed a filament surface with a small roughness (Figure 6A.2). Already, the PCL/HANp scaffold shows a flat filament surface with microporosities with a homogeneous dispersion in the matrix. It is also possible to distinguish small particles that could be hydroxyapatite nanoparticles (red arrows in Figure 6B.2). Figure 6C.2 displays a filament surface with a plasticised appearance and some roughness, which is in line with the production of this scaffold as it was submerged in PEGDA. The hydrogel appears to be a well-distributed and uniform layer, although it appears to have small irregularities.

Additionally, EDX analysis was performed to determine the presence of individual elements and the calcium/phosphate molar ratio (Table 2).

To demonstrate the homogeneous dispersion of HANp (composed of calcium and phosphate, as previously mentioned) in the matrix, the PCL/HANp and PCL/HANp/PEGDA groups were observed by EDX Si-mapping analysis, and the result is shown in Figure 7.

Regarding mechanical behaviour, compressive modulus (MPa) is reported in Figure 8 and Table 3. The mechanical behaviour is mainly conditioned by the structural characteristics, such as pore size, pore wall and connection between pores. Although with no statistical differences, the PCL/HANp group presented a higher compression modulus compared to the PCL group. In contrast, the scaffolds with PEGDA showed a slight decrease in compression behaviour compared to the same group. Despite these results, the mechanical response of the three groups presented no statistically significant differences.

### 3.3. In Vitro Cytocompatibility Test

According to ISO 10993-5:2009 guidelines, the viability was determined using PrestoBlue^TM^ on PCL (gold standard), PCL/HANP and PCL/HANP/PEGDA scaffolds in the presence of hDPSCs. The control group with the absence of scaffolds was also considered. Figure 9 and Table 4 represent the corrected absorbance values for each time point: 24, 72, 120 and 168 h. Furthermore, the statistical differences identified between the experimental groups at each time point are shown in Table 5.

The analysis results demonstrated in all groups a normal cell proliferation and growth rate until 120 h, followed by a pronounced decrease in cell viability at 168 h. Although the scaffolds with PCL/HANP presented slightly higher cell viability rates when compared with the PCL group, only at 120 h were statistically significant differences identified. Conversely, PEGDA scaffolds demonstrated significantly higher absorbance than the standard group, suggesting evidence of induction of cell adhesion and proliferation. The PCL/HANp/PEGDA scaffolds presented, overall, a superior cytocompatibility performance when in comparison with the gold-standard PCL group.

The results of the percentage of viability inhibition, normalised to the PCL group (gold standard), are presented in Figure 10 and Table 6. In accordance with Annex 3 of the ISO 10993-5:2009 guideline, inhibition of viability when superior to 30% is considered a cytotoxic effect (represented in Figure 10 by dashed lines).

The results of the percentage of viability inhibition suggest that PCL/HANP and PCL/HANP/PEGDA scaffolds can be classified as non-cytotoxic, since the percentage of viability inhibition did not exceed the pre-established limit of 30%, according to the previously mentioned guideline.

## 4. Discussion

Additive manufacturing has emerged as an innovative approach to scaffold fabrication for regeneration of critical bone defects [61]. The most widely used addictive manufacturing technique is extrusion printing, as it prints a wide variety of biomaterials at a low cost and with good precision [62]. In the present study, this technique allowed the printing of porous, biodegradable and reproducible scaffolds, based on PCL, HANp and PEGDA, as a potential substitute for the treatment of bone defects. There are several reasons for the choice of PCL as the main component of the developed scaffolds. This thermoplastic polymer is widely used because of its biocompatibility, biodegradability, ease of processing (with a melting point between 55 and 60 °C), nontoxic degradation, and adjustable composition/structure [23,24,25,63]. Nevertheless, PCL presents some inconveniences such as low adhesion and cell proliferation and its slow degradation rate given the high degree of crystallinity and hydrophobicity [64,65]. These disadvantages can be overcome with the inclusion of specific compounds of inorganic bioactive materials, namely, calcium phosphates [66,67,68]. In this sense, hydroxyapatite nanoparticles were incorporated, thus improving the bioactivity, osteoconductivity and hydrophilicity of the scaffold. Moreover, this inorganic component also tends to enhance the mechanical properties of the material [63,69]. The amount of HANp was set at 40% of the final weight of the composite, as higher proportions formed a mixture that was excessively thick, leading to clogging of the nozzle and non-uniform printing of the filaments. The incorporation of HANp in the PCL matrix was performed by the solvent casting technique. This technique was used because it is simple and allows the control of porosity, pore size and interconnectivity [70]. After the fabrication of the scaffolds using the extrusion-based technique, they were further submerged in a PEGDA solution to promote cell adhesion, proliferation and migration of the composite [47]. Although there are already several notable works on PCL/HANp scaffolds [30,31,32,35,45,49,71,72,73,74,75,76] with promising results, in reviewing the literature, no data were found on the potential of the association of these materials with PEGDA hydrogel to be used in bone defects to improve and accelerate bone regeneration. Therefore, the purpose of this study was the production, optimisation, characterisation and in vitro evaluation of three different matrices: (i) PCL scaffolds; (ii) PCL scaffolds with added HANp; and (iii) PCL/HANp scaffolds submerged in PEGDA solution. It is relevant to highlight that all scaffolds were produced successfully and demonstrated good reproducibility (Figure 2).

Regarding the FTIR spectroscopy analysis, it was possible to detect the functional groups in the respective samples (Figure 3). The spectra of the scaffold groups presented the respective characteristic absorption bands of PCL, HANp and PEGDA and are following the literature [56,57,58,60].

The wetting of the material was analysed by the contact angle of the samples for the different groups (Figure 4). The porous structure of the scaffolds did not allow a correct measurement of the angles. For this reason, flat segments of each composite were used for more precise measurement. Regarding the results of the contact angle measurement, it can be inferred that the PCL surface had a higher contact angle, supporting a hydrophobic nature [77]. Nevertheless, the addition of HANp to PCL slightly decreased the contact angle and therefore increased the wettability of the composite surface. These results are in agreement with the literature and are explained by the hydrophilic nature of hydroxyapatite [78,79]. Meanwhile, the PCL/HANp/PEGDA group demonstrated a significantly lower contact angle than the other groups. This is explained by the hydrophilic nature of the PEGDA hydrogel [47]. The addition of HANp and PEGDA may thus enhance cell activity, namely cell adhesion and proliferation, and therefore make a composite more suitable for bone tissue engineering.

To evaluate the internal and external morphology and porosity of the scaffolds, X-ray micro-computed tomography was performed (Figure 5 and Table 1). According to the results, the three groups of scaffolds were correctly produced, presenting good geometry accuracy, filaments according to the pre-established pattern and interconnected channel networks. The porosity percentage of the scaffolds was also calculated, which is defined as the percentage of void space in a solid, which is considered a material-independent property [80]. The calculated porosities of the scaffolds were approximately 50%, which is recognised as a valid value considering the literature [45,80]. The pores of the scaffolds allow the formation of bone tissue as they facilitate the migration and proliferation of osteoblasts. These structures also allow for adequate vascularisation in the scaffolds [80]. Furthermore, the porous surfaces enhance mechanical interlocking at the interface between the scaffold and the surrounding bone, providing higher mechanical stability. The addition of PEGDA slightly decreased the porosity of the scaffolds; however, there is no evidence of statistically significant differences between the different scaffolds. Thus, it appears that regardless of the material, all structures had similar porosity values. These results support the reliability of the biomanufacturing system used in the production of the samples.

SEM micrographs performed on the scaffolds revealed structures with a well-defined internal geometry with square and interconnected pores with regular dimensions and uniform distribution (Figure 6). The pores presented a size in the order of 380 µm. Studies have revealed that larger pores translate into higher mature bone formation and promote vascularisation [81,82]. This phenomenon is explained by the formation of blood vessels that, by supplying oxygen and nutrients necessary for osteoblastic activity in the larger pores of the scaffolds, promote the formation of new bone mass [82,83]. The filaments also had a regular circular geometry approximately of 400 µm in diameter, in agreement with the needle used (400 µm). At high magnifications, a rough filament surface is visible on the PCL scaffold, in contrast to the PCL/HANp scaffold with a smoother surface, microporosities and homogeneously distributed hydroxyapatite nanoparticles. The addition of a solvent (DMF) in the solvent casting method seems to promote microporosity and thus to favour cell adhesion [32,84]. In turn, the PCL/HANp/PEGDA scaffold presents a uniform layer with small irregularities.

Concerning EDX (Table 2) quantification, the presence of a considerable amount of oxygen and carbon atoms is noticeable in all samples, which was closely related to the composition of the PCL polymer matrix [85]. Moreover, the elemental composition analysis of the PCL/HANp and PCL/HANp/PEGDA groups also indicates significant amounts of calcium and phosphorus, which are the basic elements of the hydroxyapatite ceramic (Ca_10_(PO_4_)_6_(OH)_2_) [86]. In contrast, the PCL scaffold no longer shows concentrations of these chemical elements, as would be expected. EDX Ca/P determination was performed to the fabricated scaffolds, confirming the formation of CaP in the PCL/HANp and PCL/HANp/PEGDA scaffolds. The Ca/P molar ratio measured in these groups is similar to the stoichiometric value of HAp (1.67) [87]. As illustrated in the EDX Si-mapping micrographs (Figure 7), HANp (constituted by calcium and phosphorus) seems to be homogeneously distributed in the PCL matrix in the experimental groups of PCL/HANp and PCL/HANp/PEGDA. These data are following the SEM analysis results.

The values of the compressive modulus obtained for all scaffold groups are presented in Figure 8 and Table 3. The PCL/HANp group and the PCL/HANp/PEGDA group presented slightly higher and lower compression modulus, respectively, when compared to the PCL group. These results may be explained by the higher strength and stiffness properties associated with HANp [88,89]. In turn, PEGDA is considered a hydrogel and, therefore, when incorporated into the scaffold a decrease in compression modulus would be expected [42]. Notwithstanding, the average compression modulus of the three groups did not present statistically significant differences. The mechanical behaviour is mainly conditioned by structural features, such as porosity, pore size and interconnectivity. Therefore, these results are in line with previously presented results on SEM micrographs and X-ray micro-computed tomography since the values of porosity/pore size/interconnectivity are very similar in all the produced scaffolds. Overall, the results of the mechanical behaviour demonstrated that the produced scaffolds fulfil the minimum compressive modulus required for bone graft substitutes, as it is superior to 0.5 MPa [42,90]. It should be noted that the compressive strength of the structures produced is within the range of values for human cancellous bone, from 5 to 50 MPa (depending on bone density) [91]. It was visible that the standard deviation of the compression modulus of the PCL group (0.3965 MPa) is considerably lower than the standard deviation of the other groups (PCL/HANp with 1.244 MPa and PCL/HANp/PEGDA with 0.9344 MPa). This increase in the standard deviation of the PCL/HANp and PCL/HANp/PEGDA groups may be due to the multi-composition of the scaffolds and the dispersion of the materials in the PCL matrix. While in the first group (PCL), there is only one material involved, the others are multi-composites, which may affect and cause the increase in the standard deviation.

The cell viability and proliferation of hDPSCs in contact with the scaffolds was investigated by PrestoBlue^TM^ (Figure 9 and Table 4). This test was performed according to ISO 10993-5:2009 “Biological evaluation of medical devices-Part 5-Test for in vitro cytotoxicity”. The cells used in this assay were hDPSCs, given their proven potential for osteogenic lineage and consequent suitability for bone tissue regeneration [1,92,93]. Moreover, in the literature, there are other works that successfully assess the cytocompatibility of PCL and PCL/HANp scaffolds with this cell lineage [49,94,95,96]. The viability of the control group (without scaffolds) demonstrated a normal cell proliferation and growth rate, thus confirming the validity of this assay. According to Figure 9, the absorbance increased until the 5th day of cell culture, for all experimental conditions. However, on the 7th day of culture, it is noticeable that all experimental groups presented a decreased cell viability. It can be hypothesised that cells on the 7th day are in the decline/apoptosis phase because of the competitive inhibition resulting from the excess of cells in each well. During the cell death phase, there occurs the irreversible loss of cell division capacity (cell death), which is triggered by the considerable increase in intracellular calcium concentration ([Ca^2^^+^]_i_). At 24, 72 and 120 h of incubation, the PCL/HANp group showed a higher viability value (only at 120 h *p* < 0.05) compared to the PCL scaffolds group. The fact that HANP is considered a bioactive, osteoconductive and hydrophilic material may contribute to these results [63,69]. In addition, the PrestoBlue^TM^ viability assay demonstrated that the PCL/HANp/PEGDA scaffolds showed overall superior cytocompatibility performance compared with the gold standard PCL scaffold. The improved cytocompatibility results thus tend to be associated with the PEGDA hydrogel, as it has been supported by other groups working with this same material [42,97,98,99]. In sum, the results of the in vitro evaluation led to the assumption that the addition of PEGDA promoted cell migration and proliferation. According to Figure 10 and Table 6, all experimental groups can be classified as non-cytotoxic, as the % viability inhibition did not exceed the pre-established 30% limit throughout the test, according to the ISO 10993-5:2009 guideline.

## 5. Conclusions

This study describes the production, characterisation and in vitro performance of a scaffold based on PCL, HANp and PEGDA for bone application. So far, there is a wealth of literature proposing the use of PCL as a scaffold for bone pathologies. Nevertheless, to the best of our knowledge, this is the first study incorporating PCL, HANp and PEGDA in a scaffold for the healing of critical defects.

The characterisation revealed that scaffolds were successfully produced, with well-coordinated and -positioned filaments, interconnected channels, and pores propitious to the migration and proliferation of osteoblasts and stem cells. Moreover, the scaffolds with PEGDA were demonstrated to have hydrophilic properties that can also favour cellular activity without compromising the mechanical properties of the composite. The results of the in vitro test are consistent with previously demonstrated results with a superior proliferation of hDPSCs in the PEGDA groups.

The results of this paper demonstrate that PCL/HANp/PEGDA scaffolds appear to be a potential therapeutic system in the treatment of bone fractures to accelerate and improve bone regeneration. Investigation into the performance of this system in critical bone defects is a first step in its progression towards future clinical applications.

## Figures and Tables

**Figure 1 pharmaceutics-14-02643-f001:**
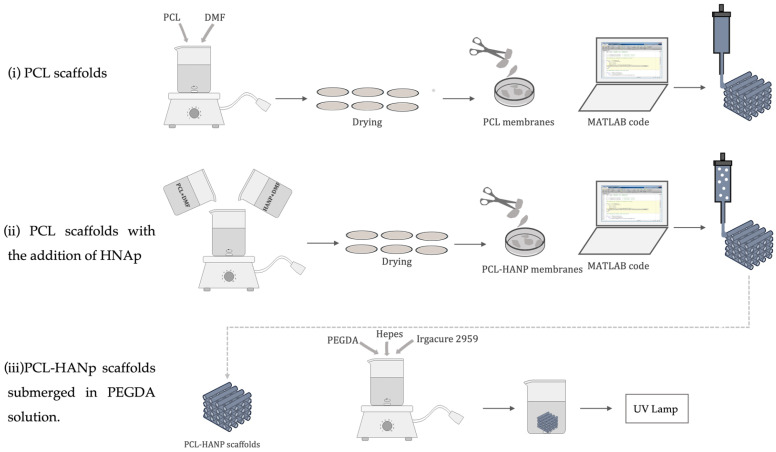
Schematic representation of the process for the manufacture of scaffolds.

**Figure 2 pharmaceutics-14-02643-f002:**
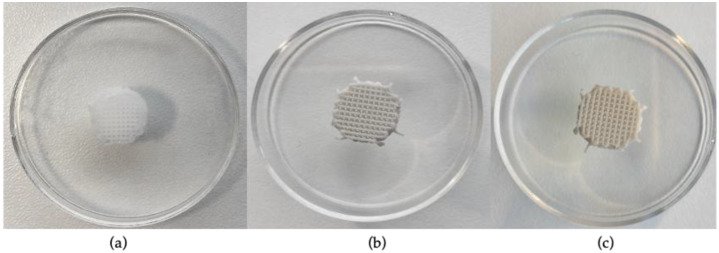
Observation of the 3D-printed scaffolds for each group: (**a**) PCL scaffolds; (**b**) PCL scaffolds with the addition of hydroxyapatite nanoparticles; and (**c**) PCL/HANp scaffolds submerged in PEGDA solution.

**Figure 3 pharmaceutics-14-02643-f003:**
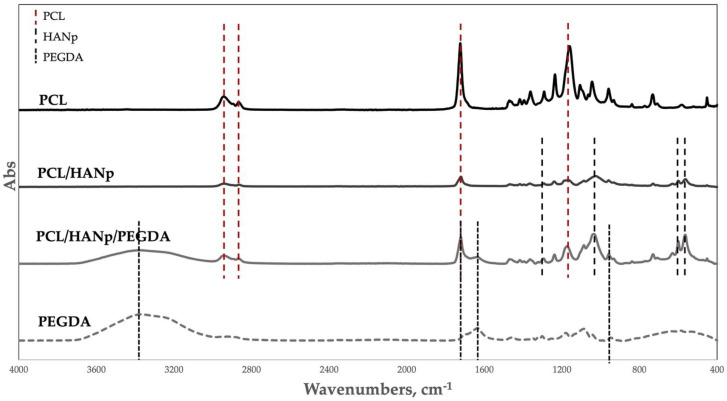
FTIR spectra of each group (PCL, PCL/HANp, PCL/HANp/PEGDA and PEGDA).

**Figure 4 pharmaceutics-14-02643-f004:**
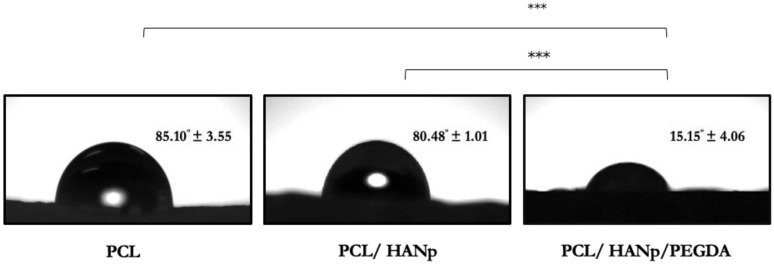
Contact angle measurements mean ± standard deviation of PCL, PCL/HANp and PCL/HANp/PEGDA scaffolds. Differences were considered statistically significant at *p* ≤ 0.05. Results’ significance is presented through the symbol (*), according to the *p* value, with three symbols, corresponding to 0.0001 < *p* ≤ 0.001.

**Figure 5 pharmaceutics-14-02643-f005:**
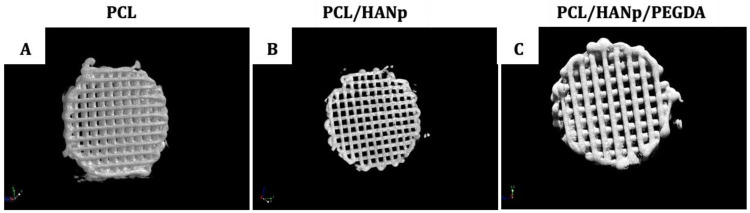
3D micro-CT images (CTVox v.3.2.0 (Bruker Micro-CT)) of (**A**) PCL scaffolds, (**B**) PCL scaffolds with the addition of hydroxyapatite nanoparticles, and (**C**) PCL/HANp scaffolds submerged in PEGDA solution.

**Figure 6 pharmaceutics-14-02643-f006:**
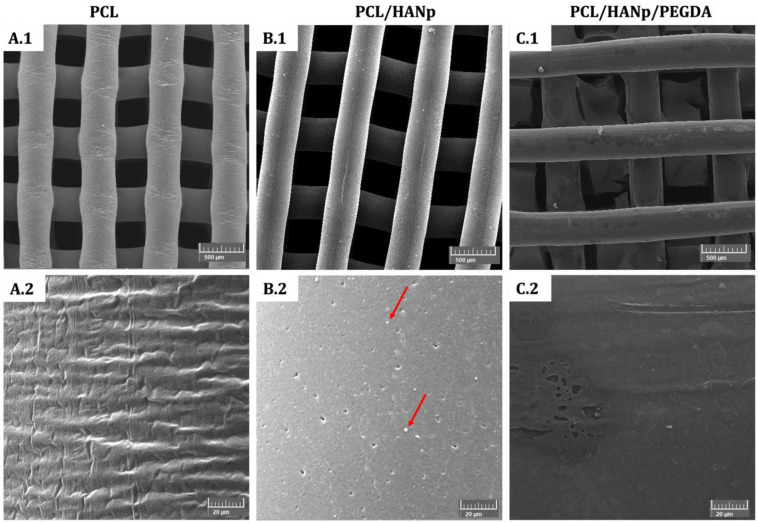
SEM images of PCL, PCL/HANp and PCL/HANp/PEGDA scaffolds. The red arrows show the presence of hydroxyapatite nanoparticles in the sample. Magnification: (**A.1**–**C.1**): 50× and (**A.2**–**C.2**): 1000×.

**Figure 7 pharmaceutics-14-02643-f007:**
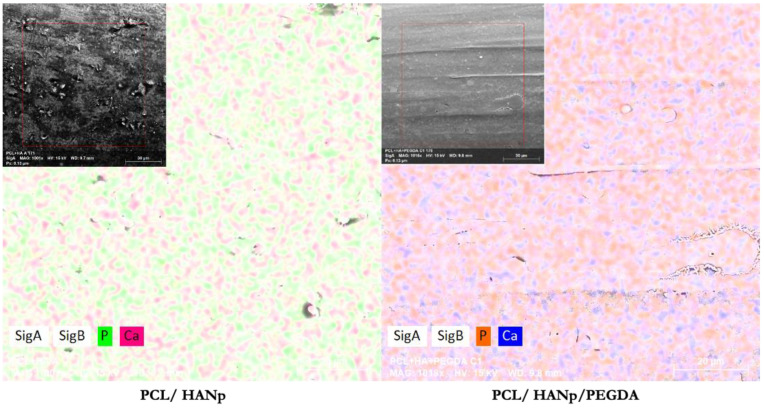
EDX Si-mapping micrographs.

**Figure 8 pharmaceutics-14-02643-f008:**
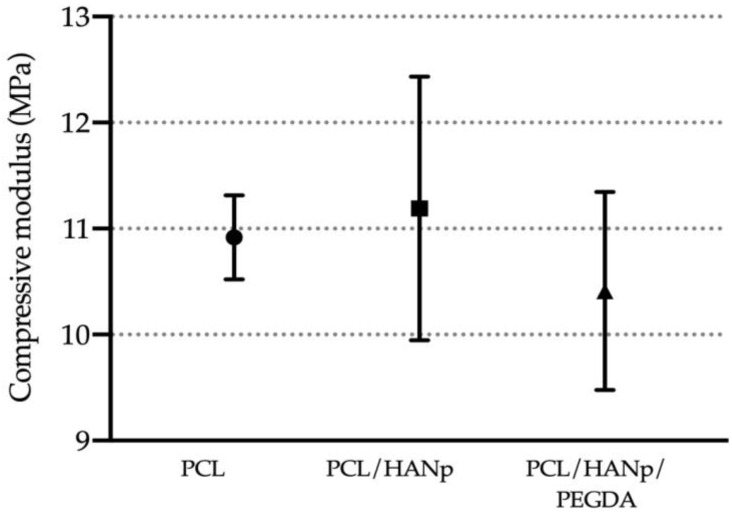
Compressive modulus (MPa) of the produced scaffolds.

**Figure 9 pharmaceutics-14-02643-f009:**
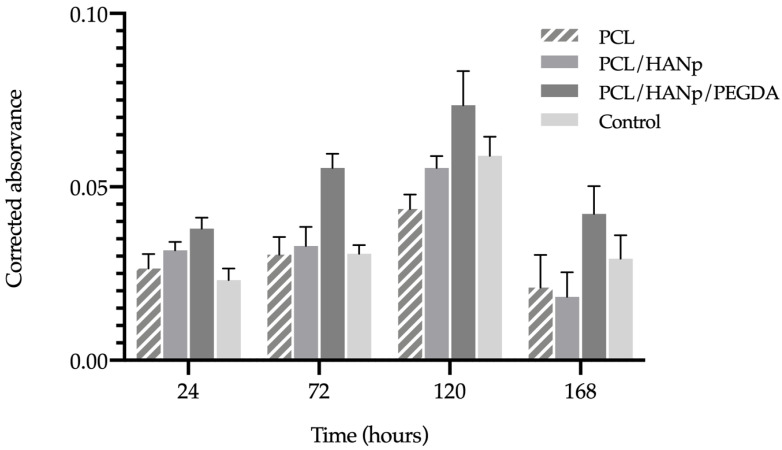
Corrected absorbance evaluated by PrestoBlue^®^ viability assay for hDPSCs after 24, 72, 120 and 168 h.

**Figure 10 pharmaceutics-14-02643-f010:**
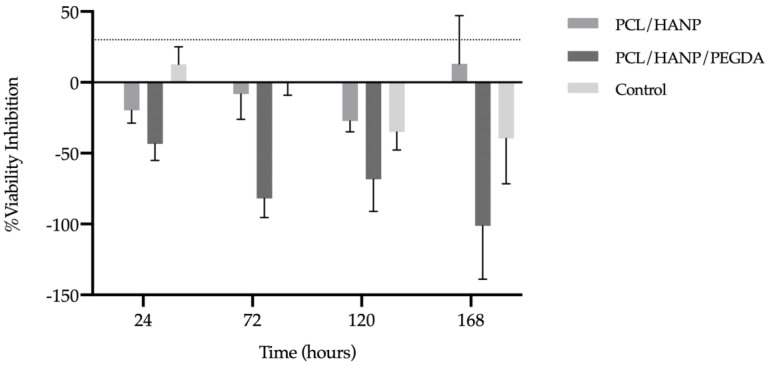
% Viability inhibition evaluated by PrestoBlue^®^ viability assay for hDPSCs after 24, 72, 120 and 168 h. Results are normalised to the PCL as 0%. The 30% threshold shown in the graph (dashed line) represents the inhibition above which the effect is considered cytotoxic (in accordance with ISO 10993-5:2009 guidelines).

**Table 1 pharmaceutics-14-02643-t001:** Porosity (CTAn v.1.20.3.0 software (Bruker Micro-CT)) of PCL, PCL/HANp and PCL/HANp/PEGDA scaffolds (mean ± standard deviation). Differences were considered statistically significant at *p* ≤ 0.05.

	PCL (1)	PCL/HANp (2)	PCL/HANp/PEGDA (3)	*p*
Porosity of scaffolds (%)	47.80 ± 0.90	52.20 ± 1.67	51.53 ± 2.00	(1) and (2)–0.051 (1) and (3)–0.065 (2) and (3)–0.052

**Table 2 pharmaceutics-14-02643-t002:** EDX analysis of the scaffolds produced and CA/P molar ratio results.

Scaffold	Oxygen (O)		Calcium (Ca)		Carbon (C)		Phosphorus (P)		Ca/P Molar Ratio
	Mass (%)	Atomic (%)	Mass (%)	Atomic (%)	Mass (%)	Atomic (%)	Mass (%)	Atomic (%)	
PCL	32.94	26.94	0.00	0.00	67.06	73.06	0.00	0.00	-
PCL/HANp	2.30	16.69	3.02	8.76	7.00	67.72	1.82	6.83	1.66
PCL/HANp/PEGDA	5.55	30.18	2.76	6.00	8.11	58.75	1.81	5.08	1.52

**Table 3 pharmaceutics-14-02643-t003:** Compressive mechanical properties of the formulations produced (mean ± sd values).

Parameter (MPa)	PCL	PCL/HANp	PCL/HANp/PEGDA
Compressive modulus E	10.92 ± 0.3965	11.19 ± 1.244	10.41 ± 0.9344

**Table 4 pharmaceutics-14-02643-t004:** Corrected absorbance evaluated by PrestoBlue^®^ viability assay for hDPSCs after 24, 72, 120 and 168 h. Results presented in mean ± SE.

	PCL	PCL/HANp	PCL/HANp/PEGDA	Control
24 h	0.027 ± 0.004	0.032 ± 0.002	0.038 ± 0.003	0.023 ± 0.003
72 h	0.031 ± 0.004	0.033 ± 0.005	0.056 ± 0.004	0.031 ± 0.002
120 h	0.044 ± 0.004	0.056 ± 0.003	0.074 ± 0.009	0.059 ± 0.005
168 h	0.021 ± 0.008	0.018 ± 0.006	0.042 ± 0.007	0.029 ± 0.006

**Table 5 pharmaceutics-14-02643-t005:** Statistical significance of viability assay for hDPSCs after 24, 72, 120 and 168 h (CT—control; ns—not significant). Results significances are presented through the symbol (*), according to the *p* value, with one, two, three or four symbols corresponding to 0.01 < *p* ≤ 0.05; 0.001 < *p* 0.01; 0.0001 < *p* ≤ 0.001 and *p* ≤ 0.0001, respectively.

	24 h				72 h				120 h				168 h			
	PCL	PCL/ HANp	PCL/HANp/ PEGDA	CT	PCL	PCL/ HANp	PCL/HANp/ PEGDA	CT	PCL	PCL/ HANp	PCL/HANp/ PEGDA	CT	PCL	PCL/ HANP	PCL/HANp/PEGDA	CT
PCL		ns	*	ns		ns	****	ns		*	****	**		ns	****	ns
PCL/HANp			ns	ns			****	ns			***	ns			****	*
PCL/HANp/PEGDA				**				****				**				*
CT																

**Table 6 pharmaceutics-14-02643-t006:** Percentage of viability inhibition evaluated by PrestoBlue^®^ viability assay for hDPSCs after 24, 72, 120 and 168 h. Results presented in mean ± SE.

	PCL/HANp	PCL/HANp/PEGDA	Control
24 h	−19.81 ± 7.72	−43.40 ± 10.16	12.72 ± 10.77
72 h	−8.20 ± 15.47	−81.97 ± 11.59	−0.82 ± 10.10
120 h	−27.22 ± 6.68	−68.48 ± 19.62	−35.13 ± 10.87
168 h	13.10 ± 29.43	−101.19 ± 32.71	−39.52 ± 27.76

## Data Availability

Further data on the reported results are available from the corresponding authors on request.

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
