# Peer review of "Assessment of 3D-Printed Polycaprolactone, Hydroxyapatite Nanoparticles and Diacrylate Poly(ethylene glycol) Scaffolds for Bone Regeneration"

_pharmaceutics, 2022, doi:10.3390/pharmaceutics14122643_

Round 1

Reviewer 1 Report

The article "Assessment of 3D printed polycaprolactone, hydroxyapatite nanoparticles and polyethylene glycol diacrylate scaffolds for bone regeneration" concerns a current problem of regenerative medicine - regeneration of bone tissue defects. One of the most actively researched and developed areas of tissue engineering is the creation of three-dimensional biostructures/scaffolds for the proliferation of cell structures using 3D printing methods. This article presents a comparative study of scaffolds based on (1) polycaprolactone (PCL), (2) PCL and hydroxyapatite nanoparticles (HANp), (3) PCL, HANp and polyethylene glycol diacrylate (PEGDA). The article could be improved by considering the following recommendations and remarks:

1.      In line 36 the word "presenting" is repeated twice, it is worth rephrasing the sentence.

2.      The introduction should be supplemented with more relevant and up-to-date research for a better disclosure of the subject. The research trend of recent years in tissue engineering is the addition of nanoparticles (most often gold or carbon) to develop a scaffold with optimal required characteristics. For example: https://doi.org/10.3390/app11178036, https://doi.org/10.3390/ijms21176163 details the applications of functional nanomaterials based on carbon nanotubes for 3D bioconstructions.

3.      The inscriptions in Figure 1 are too small - please enlarge them.

4.      Specify the concentration of PCL in DMF solution.

5.      Check that the city and country of manufacture is listed for all equipment used.

6.      In Figure 4, with the contact angle seems to be considerably larger than its numerical value (15 degrees). Explain this difference.

7.      Figure 10 is rather difficult to understand. Perhaps it would be better to give actual numerical values instead of % viability inhibition relative to PCL.

8.      Please clarify what was the control group in the in vitro compatibility test.

9.      Several references do not have a doi, please add.

Author Response

Response to the reviewers
We would like to thank the reviewer´s comments on the manuscript “Assessment of 3D printed polycaprolactone, hydroxyapatite nanoparticles and polyethylene glycol diacrylate scaffolds for bone regeneration", Ref: Pharmaceutics-2022868. The manuscript has been modified and improved according to the suggestions of the reviewers. The changes made to the document are described below. All changes made and text segments appear in the manuscript highlighted in yellow.
Reviewers' comments:
Reviewer #1: 
The article "Assessment of 3D printed polycaprolactone, hydroxyapatite nanoparticles and polyethylene glycol diacrylate scaffolds for bone regeneration" concerns a current problem of regenerative medicine - regeneration of bone tissue defects. One of the most actively researched and developed areas of tissue engineering is the creation of three-dimensional biostructures/scaffolds for the proliferation of cell structures using 3D printing methods. This article presents a comparative study of scaffolds based on (1) polycaprolactone (PCL), (2) PCL and hydroxyapatite nanoparticles (HANp), (3) PCL, HANp and polyethylene glycol diacrylate (PEGDA). The article could be improved by considering the following recommendations and remarks:
1. In line 36 the word "presenting" is repeated twice, it is worth rephrasing the sentence.
The authors appreciate the comment and have removed the word "presenting" (line 35).
2. The introduction should be supplemented with more relevant and up-to-date research for a better disclosure of the subject. The research trend of recent years in tissue engineering is the addition of nanoparticles (most often gold or carbon) to develop a scaffold with optimal required characteristics. For example: https://doi.org/10.3390/app11178036, https://doi.org/10.3390/ijms21176163 details the applications of functional nanomaterials based on carbon nanotubes for 3D bioconstructions.
Considering the Reviewer's constructive comments, the authors propose to add more relevant and up-to-date research for better dissemination of the subject as the following: 
(Line 120) “In recent years, nanoparticles (e.g., hydroxyapatite, gold, carbon) have been the subject of research as they control the structure of the material at the nanoscale. The materials show a higher resolution of the nanocomposite structure the smaller the size of these nanoparticles [1,2].”
(Line 143) “Several studies combined PCL and HANp in scaffolds due to their properties and achieved good results inherent to bone regeneration both in vitro and in vivo [32-41]. In work by Song and colleagues the results indicated that cell activities in PCL-HANp scaffolds are higher than in PCL/Hydroxyapatite, possibly due to higher hydrophilicity and porosity of the PCL-N/HA scaffold than in PCL/Hydroxyapatite [32]. Chuenjitkunt-aworn and colleagues demonstrated that PCL/HANp scaffolds exhibit higher levels of calcium deposition compared to PCL alone, and that they support the growth of various mesenchymal stem cell types [40]. Furthermore, El-Habashy et al. evaluated biopoly-mer-based hydrogel scaffolds enhanced with bioactive hybrid hydroxyapatite/polycaprolactone nanoparticles in rabbit tibial bone defects. The results demonstrated that the produced scaffolds supported bone regeneration in vivo, provided adequate biodegradation, biocompatibility and osteogenic/osteoconductive properties [41]. Previous studies also consider diacrylate poly (ethylene glycol) (PEGDA) hydrogel as an effective biomaterial in bone regeneration due to its properties such as strength, photo-crosslinkable, gelation process, hydrophilicity, and cell adhesion [42-47]. Kotturi and colleagues, assessed the physical, mechanical, and biological properties of the PEDGA-PCL scaffold toward tissue engineering applications. The results demonstrated efficacy in the combination of PCL and PEGDA showing that these scaffolds absorb nutrients over time and can provide an optimal environment for cell survival, adhesion, proliferation and migration [45]. In the study by Liu et al, a mineralised PEGDA/HA hydrogel loaded with Exendin4 (a stable analogue of the gut hormone GLP-1) was produced for the healing process of bone defects, demonstrating good biocompatibility and mechanical properties [42].”

3.      The inscriptions in Figure 1 are too small - please enlarge them.
The authors acknowledge the reviewer's comment and have enlarged the text in Figure 1.

Figure 1. Schematic representation of the process for the manufacture of scaffolds.

4.    Specify the concentration of PCL in DMF solution.
The authors acknowledge the reviewer's comment and have specified the concentration of DMF in PCL accordingly: 
(Line 330) “The PCL was dissolved in DMF (2 mL of DMF for each 0.5g of PCL) at 80°C and dried in a controlled environment for 96 hours.”
5.     Check that the city and country of manufacture is listed for all equipment used. 
The authors agree with the reviewer's suggestion and propose the following:

(Line 357 To extract qualitative chemical information, samples were analysed using the Bruker Alpha-P ATR FTIR spectrometer (Bruker, Kontich, Belgium) and Opus software.

(Line 364) The wettability of the formulations was evaluated by static contact angle meas-urement at 10s on a Theta Lite optical tensiometer (Attension, Biolin Scientific, Espoo, Finland). 

(Line 369) Micro-CT scans of the scaffolds were performed using a SkyScan mi-crotomograph model 1174, Brucker (Kontich, Belgium). Scan parameters selection, flat field correction and every operator’s choice during the scan, reconstruction and analysis steps are very important to get the best results. The scan parameters selected for the digitalizations involved in this work were: For PCL only: 50 Kv; 800 µA; 19.6 image pixel size; 4500 ms exposure; averaging frames 3; 0.9 rotation degree; no filter. For PCLHANp and PCL/HANp /PEGDA: 50 Kv; 800 µA; 19.6 image pixel size; 6500 ms exposure; averaging frames 3; 0.9 rotation degree; 0.25 Al filter (to increase the photon’s energy of the beam because of HANp presence). The scan duration was 01:10 h. Acquired radiographs were collected on a 1.3Mp CCD Charge Coupled Device (CCD) coupled to scintillator by lens. The images were then mathematically reconstructed into slices with NRecon v.1.7.0.4 software (Bruker Micro-CT) using 25% of beam hardening correction, ring artefact correction of 5 and similar contrast limits among similar specimens. A dataset composed of 573 cross-sections was obtained and a Region of Interest (ROI) was then selected over a representative amount of 300 slices; the ROI was used for the morphometry analysis made with the help of CTAn v.1.20.3.0 software (Bruker Micro-CT). CTVox v.3.2.0 (Bruker Micro-CT) was the software used to preform 3D volume rendering of the total dataset, providing a 3D viewing environment and 3D images; parameters for light and opacity control were selected.”

(Line 390) To analyse the filament and pore morphology, scaffolds from each experimental group were analysed by SEM using a Vega3 Tescan equipment (Tescan, Brno, Czechia), operating at an accelerating voltage of 20 kV, at variable magnifications and with a working distance of around 10 mm. The samples were fixed on a brass stub us-ing double-sided tape and then made electrically conductive by coating with gold/palladium (Au/Pd) thin film, by sputtering, using the sputter coater equipment for 45 seconds with 5 cm of distance between the target and the sample and 20 mA (SC7620 Quorum Technologies, Lewes, U.K.). The samples were also analysed using EDX (Xflash 6|30 from Brucker, Billerica, Massachusetts, U.S.A.).

(line 401) The tests were conducted according to ASTM STP 1173 standards, using a TA.XTplusC, (Stable Micro Systems, Godalming, U.K.)  with using an extension rate of 0.6 mm/min.
6.    In Figure 4, with the contact angle seems to be considerably larger than its numerical value (15 degrees). Explain this difference.
The authors acknowledge the reviewer's comment. The original image of the contact angle measured on a sample of PCL/HANp and PEGDA from Figure 4 is shown below (Figure 1-A). Although not clear from the image, the measurement shown in this group is 15.15° ± 4.06°. Other samples from the same group are also shown in the following figure to support this average (Figure 1 - B and C). 

Figure 1. Contact angle measurements of PCL/HANp and PEGDA scaffolds.
However, with the formatting and cropping of the image, it is possible that it has been flattened, thus leading it to appear to have a higher contact angle. Therefore, the schematic has been resized (shown in the following Figure).

Figure 4. Contact angle measurements mean ± standard deviation of PCL, PCL/HANp and PCL/HANp/PEGDA scaffolds. 
7.      Figure 10 is rather difficult to understand. Perhaps it would be better to give actual numerical values instead of % viability inhibition relative to PCL
The authors acknowledge the reviewer's suggestion. Nevertheless, the current numerical values of absorbance are already demonstrated in Figure 9 and Table 5.  The main purpose of the percentage of viability inhibition (Figure 10 and Table 6), normalized to the PCL group, is to determine whether, according to Annex 3 of ISO 10993-5:2009 guideline, the biomaterial can be considered toxic or non-toxic in the cellular environment. When viability inhibition is higher than 30% the material is considered to have a cytotoxic effect. This is intended as a supplementary analysis to the cell viability test presented in Figure 9 and Table 4.
8.      Please clarify what was the control group in the in vitro compatibility test.
The authors acknowledge and agree with the Reviewer's comment and therefore the control group has been clarified as follows:
(Line 655) “The control group with the absence of scaffolds was also considered.”
9.      Several references do not have a doi, please add. 
The authors agree with the reviewer's suggestion and have completed all references with the respective doi.

Reviewer 2 Report

The submission "Assessment of 3D printed polycaprolactone, hydroxyapatite nanoparticles and polyethylene glycol diacrylate scaffolds for bone regeneration" by Sousa et al. reports a systematic characterization and assessment of three biodegradable 3D printed scaffold cases. While the overall evaluation does not indicate a signal for preferring a particular case, the scaffolds based on polyethylene glycol diacrylate (PEGDA) demonstrated hydrophilic properties that can improve cellular activity.

The submission adequately describes the Materials and Methods, and the presentation of Results is consistent. Although the Discussion is proper, I found it a bit self-repeating at the opening of each paragraph.

However, there are two issues that the authors should address before publication in pharmaceutics:

a)   There is ineffective management of References at particular points. The authors can significantly reduce the total References number by avoiding unnecessary citations or even without apparent reasons to refer.

b)   In the discussion of the compressibility, one would expect to see a comment on the 11% deviation of PCL/HANp samples compared to the 4% of the PCL (Figure 8). If the samples are so similar (Figure 6) and the average value of the compressive modulus so "statistically insignificant" between different groups (Lines 323 & 327), then why the deviation for PCL/HANp is so high (within the same group)?

All essential points and additional comments for improvement are highlighted in the edited PDF by the Reviewer.

Author Response

Response to the reviewers

We would like to thank the reviewer´s comments on the manuscript “Assessment of 3D printed polycaprolactone, hydroxyapatite nanoparticles and polyethylene glycol diacrylate scaffolds for bone regeneration", Ref: Pharmaceutics-2022868. The manuscript has been modified and improved according to the suggestions of the reviewers. The changes made to the document are described below. All changes made and text segments appear in the manuscript highlighted in yellow.

Reviewer:

The submission "Assessment of 3D printed polycaprolactone, hydroxyapatite nanoparticles and polyethylene glycol diacrylate scaffolds for bone regeneration" by Sousa et al. reports a systematic characterization and assessment of three biodegradable 3D printed scaffold cases. While the overall evaluation does not indicate a signal for preferring a particular case, the scaffolds based on polyethylene glycol diacrylate (PEGDA) demonstrated hydrophilic properties that can improve cellular activity.

The submission adequately describes the Materials and Methods, and the presentation of Results is consistent. Although the Discussion is proper, I found it a bit self-repeating at the opening of each paragraph.

However, there are two issues that the authors should address before publication in pharmaceutics:

  1. a)   There is ineffective management of References at particular points. The authors can significantly reduce the total References number by avoiding unnecessary citations or even without apparent reasons to refer.

The authors acknowledge and agree with the Reviewer's comment and therefore the following references have been removed:

“10. Kohn, J. New approaches to biomaterials design. Nature materials 2004, 3, 745-747.

  1. Brown, B.N.; Valentin, J.E.; Stewart-Akers, A.M.; McCabe, G.P.; Badylak, S.F. Macrophage phenotype and remodeling outcomes in response to biologic scaffolds with and without a cellular component. Biomaterials 2009, 30, 1482-1491.
  2. Porter, J.R.; Ruckh, T.T.; Popat, K.C. Bone tissue engineering: a review in bone biomimetics and drug delivery strategies. Biotechnol Prog 2009, 25, 1539-1560, doi:10.1002/btpr.246.
  3. Zimmerling, A.; Yazdanpanah, Z.; Cooper, D.M.L.; Johnston, J.D.; Chen, X. 3D printing PCL/nHA bone scaffolds: exploring the influence of material synthesis techniques. Biomaterials Research 2021, 25, 3, doi:10.1186/s40824-021-00204-y.
  4. Zhang, S.; Liu, C. Research progress in osteogenic differentiation of adipose-derived stem cells induced by bioscaffold materials. Chinese Journal of Tissue Engineering Research 2020, 24, 1107.
  5. Domingos, M.; Intranuovo, F.; Russo, T.; De Santis, R.; Gloria, A.; Ambrosio, L.; Ciurana, J.; Bartolo, P. The first systematic analysis of 3D rapid prototyped poly(ε-caprolactone) scaffolds manufactured through BioCell printing: the effect of pore size and geometry on compressive mechanical behaviour and in vitro hMSC viability. Biofabrication 2013, 5, 045004, 687 doi:10.1088/1758-5082/5/4/045004.
  6. Domingos, M.; Intranuovo, F.; Gloria, A.; Gristina, R.; Ambrosio, L.; Bártolo, P.J.; Favia, P. Improved osteoblast cell affinity on plasma-modified 3-D extruded PCL scaffolds. Acta Biomaterialia 2013, 9, 5997-6005, doi:https://doi.org/10.1016/j.actbio.2012.12.031.
  7. Bartolo, P.; Domingos, M.; Gloria, A.; Ciurana, J. BioCell Printing: Integrated automated assembly system for tissue engineering constructs. CIRP Annals 2011, 60, 271-274, doi:https://doi.org/10.1016/j.cirp.2011.03.116.
  8. Rohner, D.; Hutmacher, D.W.; Cheng, T.K.; Oberholzer, M.; Hammer, B. In vivo efficacy of bone-marrow-coated polycaprolactone scaffolds for the reconstruction of orbital defects in the pig. J Biomed Mater Res B Appl Biomater 2003, 66, 574-580, doi:10.1002/jbm.b.10037.
  9. Williams, J.M.; Adewunmi, A.; Schek, R.M.; Flanagan, C.L.; Krebsbach, P.H.; Feinberg, S.E.; Hollister, S.J.; Das, S. Bone tissue engineering using polycaprolactone scaffolds fabricated via selective laser sintering. Biomaterials 2005, 26, 4817-4827, doi:10.1016/j.biomaterials.2004.11.057.
  10. Ba Linh, N.T.; Min, Y.K.; Lee, B.T. Hybrid hydroxyapatite nanoparticles-loaded PCL/GE blend fibers for bone tissue engineering. J Biomater Sci Polym Ed 2013, 24, 520-538, doi:10.1080/09205063.2012.697696.
  11. Ufere, S.K.J.; Sultana, N. Contact angle, conductivity and mechanical properties of polycaprolactone/hydroxyapatite/polypyrrole scaffolds using freeze-drying technique. ARPN J. Eng. Appl. Sci 2016, 11, 13686.
  12. Yang, X.; Yang, F.; Walboomers, X.F.; Bian, Z.; Fan, M.; Jansen, J.A. The performance of dental pulp stem cells on nanofibrous PCL/gelatin/nHA scaffolds. J Biomed Mater Res A 2010, 93, 247-257, doi:10.1002/jbm.a.32535.
  13. Aghazadeh, M.; Samiei, M.; Hokmabad, V.R.; Alizadeh, E.; Jabbari, N.; Seifalian, A.; Salehi, R. The Effect of Melanocyte Stimulating Hormone and Hydroxyapatite on Osteogenesis in Pulp Stem Cells of Human Teeth Transferred into Polyester Scaffolds. Fibers and Polymers 2018, 19, 2245-2253, doi:10.1007/s12221-018-8309-6.”

  1. b)   In the discussion of the compressibility, one would expect to see a comment on the 11% deviation of PCL/HANp samples compared to the 4% of the PCL (Figure 8). If the samples are so similar (Figure 6) and the average value of the compressive modulus so "statistically insignificant" between different groups (Lines 323 & 327), then why the deviation for PCL/HANp is so high (within the same group)?

The authors acknowledge and agree with the Reviewer's comment.

In fact, the standard deviation of PCL (0.3965 MPa) is considerably lower than the standard deviation of the other groups (PCL/HANp with 1.244 MPa and PCL/HANp/PEGDA with 0.9344 MPa).  There are different characteristics that can influence the mechanical behavior, namely porosity, interconnectivity, pore size distribution and interfacial bonding strength. Although the structures are all similar, regarding the mentioned parameters, a minimum variability among the samples is inevitable. The variability of properties can condition the mechanical behavior of the samples. Another factor that can influence and increase the standard deviation of the group is the multicomposition of the scaffolds and the dispersion of the materials in the PCL matrix. While in the first group (standard deviation: 0.3965 MPa) there is only one material involved, the others are multicomposites (standard deviation: 1.244 MPa and 0.9344) which may lead to increase in the standard deviation. However, this does not mean that the replicates in each group show significant differences between them. In the study by Li et al. the compression modulus of scaffolds composed of PCL and PCL with increasing amounts of HANp was calculated. The results showed that scaffolds with only one material (PCL group) had a compression modulus of 6.54 with a standard deviation of 0.37 MPa, while with increasing HANp in the scaffold, the modulus increased to 16.51 with a standard deviation of 2.11 MPa [1]. In similarity to the present study, the multicomposition of the scaffold seems to lead to an increase in the standard deviation of the compression modulus.

In accordance with the reviewer's comment, the following explanation has been added to the manuscript:

(Line 808)” It was visible that the standard deviation of the compression modulus of the PCL group (0.3965 MPa) is considerably lower than the standard deviation of the other groups (PCL/HANp with 1.244 MPa and PCL/HANp/PEGDA with 0.9344 MPa).  This increase in the standard deviation of the PCL/HANp and PCL/HANp/PEGDA groups may be due to the multi-composition of the scaffolds and the dispersion of the materials in the PCL matrix. While in the first group (PCL) there is only one material involved, the others are multicomposites which may affect and cause the increase of the standard deviation.”

[1] Li, Y.; Yu, Z.; Ai, F.; Wu, C.; Zhou, K.; Cao, C.; Li, W. Characterization and evaluation of polycaprolactone/hydroxyapatite composite scaffolds with extra surface morphology by cryogenic printing for bone tissue engineering. Materials & Design 2021, 205, 109712, doi:https://doi.org/10.1016/j.matdes.2021.109712.

Reviewer 3 Report

The paper is very interesting and well-prepared. It is definitely worth considering for publication. However, some minor revisions are suggested - all of them are provided in more detail below:

1) The notation of polymers should be corrected, e.g. in the title of the paper "polyethylene glycol diacrylate" should be replaced by "diacrylate poly(ethylene glycol)".

2) In second paragraph of the Introduction section Authors wrote "Several studies combined PCL and HANp in scaffolds due to their properties and achieved good results inherent to bone regeneration both in vitro and in vivo [22-31]". This information should be significantly extended considering the fact that  Author used 10 reference reports.

3) Quality of Figure 1. should be improved.

4) Section 2.3.1.: the notation of the spectral range should be corrected and written as "4000–400 cm-1" instead of "400–4000 cm-1".

5) The standards (e.g. ASTM STP 1173 standards) should be given with adequate literature reports.

Author Response

Response to the reviewers

We would like to thank the reviewer´s comments on the manuscript “Assessment of 3D printed polycaprolactone, hydroxyapatite nanoparticles and polyethylene glycol diacrylate scaffolds for bone regeneration", Ref: Pharmaceutics-2022868. The manuscript has been modified and improved according to the suggestions of the reviewers. The changes made to the document are described below. All changes made and text segments appear in the manuscript highlighted in yellow.

Reviewer :

The paper is very interesting and well-prepared. It is definitely worth considering for publication. However, some minor revisions are suggested - all of them are provided in more detail below:

1) The notation of polymers should be corrected, e.g. in the title of the paper "polyethylene glycol diacrylate" should be replaced by "diacrylate poly(ethylene glycol)".

The authors acknowledge the reviewer’s suggestion and accordingly replaced “polyethylene glycol diacrylate” for “diacrylate poly(ethylene glycol)” along the whole manuscript.

2) In second paragraph of the Introduction section Authors wrote "Several studies combined PCL and HANp in scaffolds due to their properties and achieved good results inherent to bone regeneration both in vitro and in vivo [22-31]". This information should be significantly extended considering the fact that author used 10 reference reports.

Considering the Reviewer's constructive comments, the authors propose to add more information of the references with the description of some studies:

(Line 143) “Several studies combined PCL and HANp in scaffolds due to their properties and achieved good results inherent to bone regeneration both in vitro and in vivo [29-38]. In work by Song and colleagues the results indicated that cell activities in PCL-HANp scaffolds are higher than in PCL/Hydroxyapatite, possibly due to higher hydrophilicity and porosity of the PCL-N/HA scaffold than in PCL/Hydroxyapatite [29]. Chuenjitkunt-aworn and colleagues demonstrated that PCL/HANp scaffolds exhibit higher levels of calcium deposition compared to PCL alone, and that they support the growth of various mesenchymal stem cell types [37]. Furthermore, El-Habashy et al. evaluated biopoly-mer-based hydrogel scaffolds enhanced with bioactive hybrid hydroxyap-atite/polycaprolactone nanoparticles in rabbit tibial bone defects. The results demonstrated that the produced scaffolds supported bone regeneration in vivo, provided adequate biodegradation, biocompatibility and osteogenic/osteoconductive properties [38].”

3) Quality of Figure 1. should be improved.

The authors acknowledge the reviewer's comment and have improved the quality of Figure 1.

Figure 1. Schematic representation of the process for the manufacture of scaffolds.

4) Section 2.3.1.: the notation of the spectral range should be corrected and written as "4000–400 cm-1" instead of "400–4000 cm-1".

The authors agree with the Reviewer comment, corrections were introduced to page 5, line 355, and “400–4000 cm-1” was replaced by “4000–400 cm-1”.

5) The standards (e.g. ASTM STP 1173 standards) should be given with adequate literature reports.

The authors agree with the reviewer's comment and therefore the respective ASTM literature reference has been added (Reference [54] line 298).

Author Response

Response to the reviewers

We would like to thank the reviewer´s comments on the manuscript “Assessment of 3D printed polycaprolactone, hydroxyapatite nanoparticles and polyethylene glycol diacrylate scaffolds for bone regeneration", Ref: Pharmaceutics-2022868. The manuscript has been modified and improved according to the suggestions of the reviewers. The changes made to the document are described below. All changes made and text segments appear in the manuscript highlighted in yellow.

Reviewer :

The authors were able to demonstrate the extrusion-based 3D printing technique for fabricating 3D-printed scaffolds utilizing thermoplastic polymer alone or in combinations. The methodologies and experiments seem well-planned, and the findings seem to support the stated claims. Even though the authors' findings have been adequately presented, there are a few flaws that should be addressed before the work is considered for publication. Overall, the research paper requires major revision, and the authors are encouraged to make the necessary changes before it can be considered for publication.

Abstract:

  1. Please check whether it should be 3D scaffolds or 3D printed scaffolds.

The authors agree with the Reviewer comment, corrections were introduced to page 1, line 26, and “3D scaffolds” was replaced by “3d printed scaffolds”.

  1. (Line 35-38) Please remove the following word “presenting hydrophilic properties” and please use consistent formatting for headings, units (line 37, 132), and so on throughout the manuscript.

The authors appreciate the comment and have removed the word "presenting" (line 35). Furthermore, the formatting of units forward throughout the manuscript has also been corrected.

  1. (Line 30) “thermoplastic polymer”; Please check whether it should be thermoplastic polymers?

The authors acknowledge the reviewer's question and have considered this point. The thermoplastics are considered polymers that can be softened and melted through heat [1]. These can be processed in the heat softened state or in the liquid state (e.g., by extrusion). These types of polymers can be continuously processed through the application of heat and can be recycled for the manufacture of new products. According to the literature, polycaprolactone is a semi-crystalline thermoplastic polyester [2]. It is considered thermoplastic since it is rigid at room temperature but softens and becomes malleable after being heated to 60°C. When the material cools down to room temperature, its rigidity is restored [3].

[1] Mallick, P.K. 5 - Thermoplastics and thermoplastic–matrix composites for lightweight automotive structures. In Materials, Design and Manufacturing for Lightweight Vehicles, Mallick, P.K., Ed.; Woodhead Publishing: 2010; pp. 174-207.

[2] Ragaert, K.; Baere, I.D.; Cardon, L.; Degrieck, J. Bulk mechanical properties of thermoplastic poly-e-caprolactone. 2014.

[3] Aguilar SM, Shea JD, Al-Joumayly MA, Van Veen BD, Behdad N, Hagness SC. Dielectric characterization of PCL-based thermoplastic materials for microwave diagnostic and therapeutic applications. IEEE Trans Biomed Eng. (2012) Mar;59(3):627-33. doi: 10.1109/TBME.2011.2157918.

  1. The abstract includes more background and method details, but only a few results are reported. In the abstract, the authors ought to emphasize a few significant findings.

Considering the constructive comments of the Reviewer, the authors propose the following changes to the abstract in order to emphasize more significant findings:

(Line 33) “Through the findings it was possible to conclude that in all groups, the scaffolds were successfully produced presenting networks of interconnected channels, adequate porosity for migration and proliferation of osteoblasts (approximately 50%). Furthermore, according to the in vitro analysis, all groups were considered non-cytotoxic in contact with the cells. Nevertheless, the group with PEGDA revealed hydrophilic properties (15.15° ± 4.06), adequate mechanical performance (10.41MPa ± 0.934) and demonstrated significantly higher cell viability than the other groups analyses. The scaffolds with PEGDA suggested to increase cell adhesion and proliferation, thus being more appropriate for bone regeneration.”

  1. Please check the authors guideline for keywords and format it accordingly (5-6 keywords are generally accepted).

The authors acknowledge the reviewer’s comment. The keywords have been updated accordingly:

(line 45-46) “Bone regeneration; Critical Bone Defects; Hydroxyapatite nanoparticles; Polycaprolactone; Diacrylate poly(ethylene glycol); Scaffolds.”

  1. (Line 29-31) The reader will be confused by these wordy statements. Instead, the authors might arrange it in a more straightforward manner, such as "three scaffolds of various composition, namely PCL, PCL/HANp, and PCL/HANp/PEGDA, were manufactured by extrusion”.

The authors agree with the reviewer comment and propose the following to be introduced in the abstract:

(line 29-31) “For the first time, three scaffolds of various composition, namely polycaprolactone (PCL), PCL/ hydroxyapatite nanoparticles (HANp), and PCL/HANp/ diacrylate poly(ethylene glycol) (PEGDA), were manufactured by extrusion.”

Introduction & Methods:

  1. The introduction fails to highlight the novelty of the work. The thermoplastic polymers used in the study (PCL, HANp, and PEGDA) are well-established and have been commonly used either alone or in combination for bone regeneration. The authors should give emphasis showing the novelty of the study.
  2. Moreover, the introduction also lacks sufficient background. The authors should cover what has been studied so far in prior literature and what is missing and how this study can help to fill the gap.

Considering the constructive comments of the Reviewer (7. and 8.), the authors propose to add further information from the already developed studies used alone or in combination of PCL, HANp and PCL with the following paragraph:

(Line 143) “Several studies combined PCL and HANp in scaffolds due to their properties and achieved good results inherent to bone regeneration both in vitro and in vivo [29-38]. In work by Song and colleagues the results indicated that cell activities in PCL-HANp scaffolds are higher than in PCL/Hydroxyapatite, possibly due to higher hydrophilicity and porosity of the PCL-N/HA scaffold than in PCL/Hydroxyapatite [29]. Chuenjitkunt-aworn and colleagues demonstrated that PCL/HANp scaffolds exhibit higher levels of calcium deposition compared to PCL alone, and that they support the growth of various mesenchymal stem cell types [37]. Furthermore, El-Habashy et al. evaluated biopoly-mer-based hydrogel scaffolds enhanced with bioactive hybrid hydroxyapatite/polycaprolactone nanoparticles in rabbit tibial bone defects. The results demonstrated that the produced scaffolds supported bone regeneration in vivo, provided adequate biodegradation, biocompatibility and osteogenic/osteoconductive properties [38]. Previous studies also consider diacrylate poly(ethylene glycol) (PEGDA) hydrogel as an effective biomaterial in bone regeneration due to its properties such as strength, photo-crosslinkable, gelation process, hydrophilicity, and cell adhesion [39-44]. Kotturi and colleagues, assessed the physical, mechanical, and biological properties of the PEDGA-PCL scaffold toward tissue engineering applications. The results demonstrated efficacy in the combination of PCL and PEGDA showing that these scaffolds absorb nutrients over time and can provide an optimal environment for cell survival, adhesion, proliferation and migration [42]. In the study by Liu et al, a mineralised PEGDA/HA hydrogel loaded with Exendin4 (a stable analogue of the gut hormone GLP-1) was produced for the healing process of bone defects, demonstrating good biocompatibility and mechanical properties [39].”

According to the reviewer's comment, the article was also modified in order to highlight the novelty of the presented article:

(Line 189) “For the present study, combined scaffolds of PCL, HANp and PEGDA were pro-duced by an extrusion additive manufacturing system. To the best of our knowledge, this is the first study to incorporate these three materials into a scaffold for bone application. To fill the gap in the literature, the production of these compounds from the incorporation of HANp into synthetic polymers (PCL) with the coating of a photo-crosslinkable hydrogel, combines the advantages inherited by each of these components. Therefore, it is expected that, while PCL/HANp provides mechanical support, osteoconductivity and interconnectivity be-tween the pores for cell proliferation, the PEGDA coating provides more hydrophilicity to the structure and better characteristics for cell adhesion.”

At the end of the introduction, the main objective of this work was also emphasised:

(Line 203) “Thus, this study aims to lay the groundwork for future research into the use of these three materials (PCL, HANp and PEGDA) for more accelerated and effective bone regeneration.”

  1. “In the present work, a biomanufacturing system….developed by the CDRSP-IPLeiria, was used [47,50-58]”. Please check whether the sentence actually require so many refs.

The authors acknowledge and agree with the Reviewer's comment and therefore the following references have been removed:

“52. Domingos, M.; Intranuovo, F.; Russo, T.; De Santis, R.; Gloria, A.; Ambrosio, L.; Ciurana, J.; Bartolo, P. The first systematic 685 analysis of 3D rapid prototyped poly(ε-caprolactone) scaffolds manufactured through BioCell printing: the effect of pore 686 size and geometry on compressive mechanical behaviour and in vitro hMSC viability. Biofabrication 2013, 5, 045004, 687 doi:10.1088/1758-5082/5/4/045004. 688

  1. Domingos, M.; Intranuovo, F.; Gloria, A.; Gristina, R.; Ambrosio, L.; Bártolo, P.J.; Favia, P. Improved osteoblast cell affinity 689 on plasma-modified 3-D extruded PCL scaffolds. Acta Biomaterialia 2013, 9, 5997-6005, 690 doi:https://doi.org/10.1016/j.actbio.2012.12.031.
  2. Bartolo, P.; Domingos, M.; Gloria, A.; Ciurana, J. BioCell Printing: Integrated automated assembly system for tissue 692 engineering constructs. CIRP Annals 2011, 60, 271-274, doi:https://doi.org/10.1016/j.cirp.2011.03.116.”

  1. Given that the scaffolds could be made directly from hydrogel using the same semi-solid extrusion 3D printing technique, what was the scientific justification for picking the formulation composition and need for membrane preparation?

The authors acknowledge the reviewer's pertinent question.

The extrusion technique used in this study consisted of heating the material (PCL and HANp), and pneumatic and mechanical extrusion, i.e., compressed air and a rotating spindle were used for controlled deposition of the composite, respectively. For the deposition of hydrogels, the most suitable extrusion technique is pneumatic/piston, in order to maintain the material properties in the printing process. Nevertheless, concerning PEGDA, its deposition through the latter method would not be so advantageous, since, as this material presents a high fluidity, the filament does not remain uniform after printing. Furthermore, the high temperatures of PCL/HANp extrusion would lead to PEGDA degradation. This hydrogel only becomes stable and presents some consistency after UV photopolymerisation, which can only happen after the deposition of the material, and in this case, we would keep the little control and uniformity of the filament. In this sense, using 3D printing to process PEGDA is not favourable, in this case it does not add improvements in our sample, because the control in deposition and interconnectivity between pores, characteristics of samples obtained by 3D printing, are not achieved with this hydrogel. Therefore, it was chosen to perform a coating with PEGDA as this is a simple, fast method that would not lead to very high temperatures since the scaffold would already be at room temperature. This PEGDA coating provides more hydrophilicity and better characteristics for cell adhesion, while PCL/HANp provides mechanical support, osteoconductivity and interconnectivity between pores for cell proliferation.

  1. More information is needed in the method section, such as how many total slices were collected and how many layers were chosen as areas of interest and analyzed in Micro-CT imaging. How long was the acquisition time? What detector was employed, and what approach was used for picture reconstruction? etc.

The authors acknowledge the reviewer’s comment. In consideration of the reviewer's questions, each was duly answered separately.

  • For the Micro-CT analysis, a dataset consisting of 573 cross sections was collected.
  • A selected Region of Interest (ROI) was selected over a representative amount of 300 slices. The ROI was used for the morphometry analysis.
  • CTVox v.3.2.0 (Bruker micro-CT) was the software used to preform 3D volume rendering of the total dataset, providing a 3D viewing environment and 3D images; some parameters for light and opacity were selected.
  • The exposure time for PCL scaffolds was 4500 ms and for PCL/HANp and PCL/HANp/PEGDA scaffolds was 6500 ms. The scan duration was 01:10 h.
  • Regarding the detector employed, the acquired radiographs were collected on a 1.3Mp CCD Charge Coupled Device (CCD) coupled to scintillator by lens.
  • For the picture reconstruction, the images were mathematically reconstructed into slices with NRecon v.1.7.0.4 software (Bruker micro-CT) using 25% of beam hardening correction, ring artefact correction of 5 and similar contrast limits among similar specimens and CTVox v.3.2.0 (Bruker micro-CT) was the software used to preform 3D volume rendering of the total dataset, providing a 3D viewing environment and 3D images; some parameters for light and opacity were selected.

Furthermore, the authors have reformulated and added information in the Micro-CT methodology section:

(Line 369) “Micro-CT scans of the scaffolds were performed using a SkyScan mi-crotomograph model 1174, Brucker (Kontich, Belgium). Scan parameters selection, flat field correction and every operator’s choice during the scan, reconstruction and analysis steps are very important to get the best results. The scan parameters selected for the digitalizations involved in this work were: For PCL only: 50 Kv; 800 µA; 19.6 image pixel size; 4500 ms exposure; averaging frames 3; 0.9 rotation degree; no filter. For PCLHANp and PCL/HANp /PEGDA: 50 Kv; 800 µA; 19.6 image pixel size; 6500 ms exposure; averaging frames 3; 0.9 rotation degree; 0.25 Al filter (to increase the photon’s energy of the beam because of HANp presence). The scan duration was 01:10 h. Acquired radiographs were collected on a 1.3Mp CCD Charge Coupled Device (CCD) coupled to scintillator by lens. The images were then mathematically reconstructed into slices with NRecon v.1.7.0.4 software (Bruker Micro-CT) using 25% of beam hardening correction, ring artefact correction of 5 and similar contrast limits among similar specimens. A dataset composed of 573 cross-sections was obtained and a Region of Interest (ROI) was then selected over a representative amount of 300 slices; the ROI was used for the morphometry analysis made with the help of CTAn v.1.20.3.0 software (Bruker Micro-CT). CTVox v.3.2.0 (Bruker Micro-CT) was the software used to preform 3D volume rendering of the total dataset, providing a 3D viewing environment and 3D images; parameters for light and opacity control were selected.”

  1. Similarly, the authors should add more information on the method section for FTIR, SEM EDX, etc., like what was the working distance, current used, how long was sputtering process conducted, which instruments were used and their country of origin.

The authors agree with the reviewer's comment and propose the following to supplement the information in the methods section:

(Line 354) To extract qualitative chemical information, samples were analysed using the Bruker Alpha-P ATR FTIR spectrometer (Bruker, Kontich, Belgium) and Opus software.

(Line 360) The wettability of the formulations was evaluated by static contact angle meas-urement at 10s on a Theta Lite optical tensiometer (Attension, Biolin Scientific, Espoo, Finland).

(Line 386) To analyse the filament and pore morphology, scaffolds from each experimental group were analysed by SEM using a Vega3 Tescan equipment (Tescan, Brno, Czechia), operating at an accelerating voltage of 20 kV, at variable magnifications and with a working distance of around 10 mm. The samples were fixed on a brass stub us-ing double-sided tape and then made electrically conductive by coating with gold/palladium (Au/Pd) thin film, by sputtering, using the sputter coater equipment for 45 seconds with 5 cm of distance between the target and the sample and 20 mA (SC7620 Quorum Technologies, Lewes, U.K.). The samples were also analysed using EDX (Xflash 6|30 from Brucker, Billerica, Massachusetts, U.S.A.).

(Line 398) The tests were conducted according to ASTM STP 1173 standards, using a TA.XTplusC, (Stable Micro Systems, Godalming, U.K.)  with using an extension rate of 0.6 mm/min.

  1. The printing speed seems to be very high (240 mm/s). For printing scaffold of D x h = 10 x 3 mm, this should happen in few seconds.

The authors agree with the reviewer's comment, it was a clerical error. The correct speed is 240 mm/min. It has been rephrased in the manuscript to read as follows:

(Line 225) “The parameters employed were 240 mm/min of deposition velocity”

  1. Please use consistent formatting for headings, units (line 132), and so on throughout the manuscript.

The authors acknowledge the reviewer's comment and the formatting of units throughout the manuscript has been corrected.

Results, Discussion, & Conclusions:

  1. For the manufactured scaffolds using semi-solid extrusion technique, how did the authors ensure the viscosity or rheology of the three different matrices(membranes/hydrogels) was suitable for 3D-printing?

The authors acknowledge the question exposed by the reviewer.

The conditions used to produce the 3D scaffolds in PCL and PCL/HANp were adapted from other works previously published in the literature [1,2].

To obtain an ideal viscosity, the processing temperature is adjusted in order to acquire a viscosity suitable for the deposition of the material and obtaining 3D scaffolds with the desired characteristics, in terms of interconnectivity between pores and controlled porosity. The addition of PEGDA to PCL/HANp scaffolds was performed through the coating process, and 3D printing was not applied here. This means that the addition of PEGDA was performed after the printing of the PCL/HANp scaffold.

[1] Domingos, M.; Dinucci, D.; Cometa, S.; Alderighi, M.; Bártolo, P.J.; Chiellini, F. Polycaprolactone Scaffolds Fabricated via Bioextrusion for Tissue Engineering Applications. International Journal of Biomaterials 2009, 2009, 239643, doi:10.1155/2009/239643.

[2] Biscaia, S.; Branquinho, M.V.; Alvites, R.D.; Fonseca, R.; Sousa, A.C.; Pedrosa, S.S.; Caseiro, A.R.; Guedes, F.; Patrício, T.; Viana, T.; et al. 3D Printed Poly(É›-caprolactone)/Hydroxyapatite Scaffolds for Bone Tissue Engineering: A Comparative Study on a Composite Preparation by Melt Blending or Solvent Casting Techniques and the Influence of Bioceramic Content on Scaffold Propertie. Int J Mol Sci 2022, 23, 2318, doi:10.3390/ijms23042318.

  1. Since, the printing was carried out at higher temperature i.e., 85 °C, does this affect the rheology of the membrane? Furthermore, how did the authors ensure that the 3D-printed structures (at room temperature, RT) did not collapse following extrusion, as is common with semi-solid extruded products? Is there any possibility that this temperature gradient and heat transfer phenomenon would affect product quality and morphology? SARA

The authors acknowledge the reviewer's comment.

The hydrogel had no contact with these high temperatures because when it was added, the PCL/HANp samples were already solidified and at room temperature, and therefore their rheology was not altered. As mentioned in the previous answer, PEGDA was incorporated in the PCL/HANp scaffolds by the coating process, so the "collapse following extrusion" does not occur, as really expected in the 3D printing of these materials/hydrogels.

  1. The FTIR pictures are not clear enough to draw judgements. The authors should highlight the peaks of interest or modify the Y-axis to better illustrate the peaks. “PEGDA are represented at 1638 cm-1 (double peak due to elongation of the vinyl groups), and 910 cm1 and 1720 cm-1 [63]”. PEGDA's FT-IR peaks are so faint and obscure that they are difficult to distinguish.

The authors understand the suggestions raised by the Reviewer. Thus, the quality of Figure 3 (FTIR spectra of each group (PCL, PCL/HANp, PCL/HANp/PEGDA and PEGDA)) was improved for a clearer perception of each of the peaks. Furthermore, the peaks of interest of each sample were highlighted for better compression and distinction of the results. 

Figure 3. FTIR spectra of each group (PCL, PCL/HANp, PCL/HANp/PEGDA and PEGDA).

  1. Moreover, what led to the broad peak near 3600-3200 cm-1 in PEGDA. The X-axis should be “wavenumbers”.

The authors acknowledge the reviewer's comments. According to Zaharia et al. the characteristic band of OH stretching in PEGDA is between 3500 and 3400 cm-1.  Therefore, the authors propose the following to complement the FTIR information:

(Line 524)“ Finally, in the PCL/HANp/PEGDA sample, the characteristic peaks of PEGDA are represented at 1638 cm-1 and 910 cm-1 (double peak due to elongation of vinyl groups) and 1720 cm-1 . Another characteristic peak of PEGDA is the OH stretching represented between 3500 and 3400 cm-1 [59,60].

  1. Zaharia, A.; Gavrila, A.M.; Caras, I.; Trica, B.; Chiriac, A.L.; Gifu, C.I.; Neblea, I.E.; Stoica, E.B.; Dolana, S.V.; Iordache, T.V. Molecularly Imprinted Ligand-Free Nanogels for Recognizing Bee Venom-Originated Phospholipase A2 Enzyme. Polymers (Basel) 2022, 14, doi:10.3390/polym14194200.
  2. Kianfar, P.; Vitale, A.; Dalle Vacche, S.; Bongiovanni, R. Enhancing properties and water resistance of PEO-based electrospun nanofibrous membranes by photo-crosslinking. Journal of Materials Science 2021, 56, 1-18, doi:10.1007/s10853-020-05346-3.

Regarding the word "wavenumber", it has been replaced by "wavenumbers". The updated Figure 3 is represented in the answer to the previous question (17.).

  1. After the PCL/HANp and PEGDA scaffolds were prepared and cured by UV, did the authors observe any crosslinking or molecular interaction phenomenon, which is a possibility to occur after curing PEGDA using UV?

The authors acknowledge the reviewer's relevant question.

PEGDA is a biocompatible and photo-crosslinkable hydrogel widely used in tissue regeneration applications [1]. Under the presence of a photoinitiator (as Irgacure 2959, (1-[4-(2-Hydroxyethoxy)-phenyl]-2-hydroxy-2-methyl-1-propane-1-one)) and UV exposure, the double-bonded acrylate groups on PEGDA can initiate rapid photopolymerisation in order to produce a 3D polymer network [2]. The cross-linking reaction of PEGDA includes two main stages (Figure 1): (1) the photo-initiator (Irgacure 2959) is exposed to UV light and an initiator molecule is decomposed and forms two free radicals; and (2) the free radicals combine and react with a monomer, thereby initiating a PEGDA chain by opening its carbon-carbon bond. The polymer chains are propagated through reactions with available vinyl bonds that were in monomers, or in other polymer chains. The reactive monomers bond to form large molecules, which continue to grow until two chains combine and end the reaction [3].

Figure 1. Schematic representation of the process of PEGDA reaction with a photoinitiator [3].

After crosslinking PEGDA on the PCL/HANp scaffold, a more rigid structure and potentially a better-defined shape was observed.

[1] Hockaday LA, Kang KH, Colangelo NW, Cheung PY, Duan B, Malone E, Wu J, Girardi LN, Bonassar LJ, Lipson H, Chu CC, Butcher JT. Rapid 3D printing of anatomically accurate and mechanically heterogeneous aortic valve hydrogel scaffolds. Biofabrication. 2012 Sep;4(3):035005. doi: 10.1088/1758-5082/4/3/035005.  

[2] Mei Q, Rao J, Bei HP, Liu Y, Zhao X. 3D Bioprinting Photo-Crosslinkable Hydrogels for Bone and Cartilage Repair. Int J Bioprint. 2021 Jun 24;7(3):367. doi: 10.18063/ijb.v7i3.367.

[3] Yang, W.; Yu, H.; Liang, W.; Wang, Y.; Liu, L. Rapid Fabrication of Hydrogel Microstructures Using UV-Induced Projection Printing. Micromachines 2015, 6, 1903-1913. doi:10.3390/mi6121464

  1. Based on Figure 5, it appears that the scaffolds' borders are inconsistent and poorly printed, and the structure is not totally round. The authors may consider about including CAD model images and discussing what caused deviations from the model structure.

The authors understand the suggestion raised by the Reviewer regarding the borders of the scaffolds. In this study, the composites were designed using the MATLAB program.

The scaffolds were produced with the 3D printing support structure, as presented in the following Figure.

Therefore, after printing, the supports are removed, and the part is cleaned. Nevertheless, in removing the supports it is possible that there is some loss of definition of the borders of the scaffold. Despite this loss of definition, the viability of the scaffolds is not compromised. However, in a future work, it will be possible to polish the scaffolds in order to make the edges of the scaffolds more consistent and with less deviations from the designed model. 

Another factor that can compromise Micro-CT images is the challenge of distinguishing between the interphase of the material and the resin used to support the material in the analysis (Following Figure). Despite contrast enhancement as well as careful voxel size selection, the resin can compromise some definition of the scaffold borders [1].

[1] Rashidi A, Olfatbakhsh T, Crawford B, Milani AS. A Review of Current Challenges and Case Study toward Optimizing Micro-Computed X-Ray Tomography of Carbon Fabric Composites. Materials (Basel). 2020 Aug 14;13(16):3606. doi: 10.3390/ma13163606.

  1. It is understandable that the study focuses more on the preparation, mechanical tests, and in-vitro assessment of the scaffolds, the authors are encouraged to add TGA and DSC results of the thermoplastic polymers and prepared scaffolds. This would be beneficial to assess if there were any changes in the thermal properties of the polymers before and after extrusion.

The authors appreciate the reviewer's suggestion.

This study aims the production, characterisation and in vitro performance of a scaffold based on PCL, HANp and PEGDA for bone application. Concerning the thermal properties of PCL and PCL/HANp scaffolds, these have already been reported extensively in the literature [1-3]. In several studies, PCL and PCL/HANp scaffolds were evaluated by DSC thermograms, which indicated that the addition of HANp promoted a decrease in endothermic melting enthalpies due to the high crystallinity of this ceramic that can affect the crystalline properties of the polymer and accelerate the nucleation of PCL chain segments [1-3]. In the study by Biscaia et al. the TGA analysis revealed a decrease in mass loss as a function of the amount of HANp present in the sample. Moreover, it also showed thermal stability at the processing temperatures used (very similar to those of the present study) for the preparation and manufacture of scaffolds [1].

Regarding PEGDA, the addition of this hydrogel to the PCL-HANp scaffolds was performed through the coating process, i.e., the addition of PEGDA was performed after the printing of the PCL/HANp scaffold. For this reason, the study of the thermal properties of PEGDA were not performed since this material does not undergo temperature variation. This means that there was no extrusion of the hydrogel and therefore, there are no differences before and after.

[1] Biscaia, S.; Branquinho, M.V.; Alvites, R.D.; Fonseca, R.; Sousa, A.C.; Pedrosa, S.S.; Caseiro, A.R.; Guedes, F.; Patrício, T.; Viana, T.; et al. 3D Printed Poly(É›-caprolactone)/Hydroxyapatite Scaffolds for Bone Tissue Engineering: A Comparative Study on a Composite Preparation by Melt Blending or Solvent Casting Techniques and the Influence of Bioceramic Content on Scaffold Propertie. Int J Mol Sci 2022, 23, 2318, doi:10.3390/ijms23042318.

[2] Cestari, F.; Petretta, M.; Yang, Y.; Motta, A.; Grigolo, B.; Sglavo, V.M. 3D printing of PCL/nano-hydroxyapatite scaffolds derived from biogenic sources for bone tissue engineering. Sustainable Materials and Technologies 2021, 29, e00318, doi:https://doi.org/10.1016/j.susmat.2021.e00318.

[3] Koupaei, N.; Karkhaneh, A. Porous crosslinked polycaprolactone hydroxyapatite networks for bone tissue engineering. Tissue Eng Regen Med 2016, 13, 251-260, doi:10.1007/s13770-016-9061-x.

  1. (Line 385-387) “This thermoplastic polymer is widely used due to its biocompatibility, biodegradability……….and adjustable composition/structure [32-34,66-69]”. The same has already been mentioned in the introduction, and no need for additional references.

The authors acknowledge and agree with the Reviewer's comment and therefore the following references have been removed:

“ 35. Zhang, S.; Liu, C. Research progress in osteogenic differentiation of adipose-derived stem cells induced by bioscaffold materials. Chinese Journal of Tissue Engineering Research 2020, 24, 1107.

  1. Rohner, D.; Hutmacher, D.W.; Cheng, T.K.; Oberholzer, M.; Hammer, B. In vivo efficacy of bone-marrow-coated polycaprolactone scaffolds for the reconstruction of orbital defects in the pig. J Biomed Mater Res B Appl Biomater 2003, 66, 574-580, doi:10.1002/jbm.b.10037.
  2. Williams, J.M.; Adewunmi, A.; Schek, R.M.; Flanagan, C.L.; Krebsbach, P.H.; Feinberg, S.E.; Hollister, S.J.; Das, S. Bone tissue engineering using polycaprolactone scaffolds fabricated via selective laser sintering. Biomaterials 2005, 26, 4817-4827, doi:10.1016/j.biomaterials.2004.11.057.
  3. Ba Linh, N.T.; Min, Y.K.; Lee, B.T. Hybrid hydroxyapatite nanoparticles-loaded PCL/GE blend fibers for bone tissue engineering. J Biomater Sci Polym Ed 2013, 24, 520-538, doi:10.1080/09205063.2012.697696.”

  1. (Line 492-493) “Moreover, in the literature there are other works ….. scaffolds with this cell lineage [51,100-104]”. Looking over the manuscript, overall, it appears to have a large number of references that may or may not be relevant. Please include the references that are most pertinent.

The authors acknowledge and agree with the Reviewer's comment and therefore the following references have been removed:

“10. Kohn, J. New approaches to biomaterials design. Nature materials 2004, 3, 745-747.

  1. Brown, B.N.; Valentin, J.E.; Stewart-Akers, A.M.; McCabe, G.P.; Badylak, S.F. Macrophage phenotype and remodeling outcomes in response to biologic scaffolds with and without a cellular component. Biomaterials 2009, 30, 1482-1491.
  2. Porter, J.R.; Ruckh, T.T.; Popat, K.C. Bone tissue engineering: a review in bone biomimetics and drug delivery strategies. Biotechnol Prog 2009, 25, 1539-1560, doi:10.1002/btpr.246.
  3. Zimmerling, A.; Yazdanpanah, Z.; Cooper, D.M.L.; Johnston, J.D.; Chen, X. 3D printing PCL/nHA bone scaffolds: exploring the influence of material synthesis techniques. Biomaterials Research 2021, 25, 3, doi:10.1186/s40824-021-00204-y.
  4. Zhang, S.; Liu, C. Research progress in osteogenic differentiation of adipose-derived stem cells induced by bioscaffold materials. Chinese Journal of Tissue Engineering Research 2020, 24, 1107.
  5. Domingos, M.; Intranuovo, F.; Russo, T.; De Santis, R.; Gloria, A.; Ambrosio, L.; Ciurana, J.; Bartolo, P. The first systematic analysis of 3D rapid prototyped poly(ε-caprolactone) scaffolds manufactured through BioCell printing: the effect of pore size and geometry on compressive mechanical behaviour and in vitro hMSC viability. Biofabrication 2013, 5, 045004, 687 doi:10.1088/1758-5082/5/4/045004.
  6. Domingos, M.; Intranuovo, F.; Gloria, A.; Gristina, R.; Ambrosio, L.; Bártolo, P.J.; Favia, P. Improved osteoblast cell affinity on plasma-modified 3-D extruded PCL scaffolds. Acta Biomaterialia 2013, 9, 5997-6005, doi:https://doi.org/10.1016/j.actbio.2012.12.031.
  7. Bartolo, P.; Domingos, M.; Gloria, A.; Ciurana, J. BioCell Printing: Integrated automated assembly system for tissue engineering constructs. CIRP Annals 2011, 60, 271-274, doi:https://doi.org/10.1016/j.cirp.2011.03.116.
  8. Rohner, D.; Hutmacher, D.W.; Cheng, T.K.; Oberholzer, M.; Hammer, B. In vivo efficacy of bone-marrow-coated polycaprolactone scaffolds for the reconstruction of orbital defects in the pig. J Biomed Mater Res B Appl Biomater 2003, 66, 574-580, doi:10.1002/jbm.b.10037.
  9. Williams, J.M.; Adewunmi, A.; Schek, R.M.; Flanagan, C.L.; Krebsbach, P.H.; Feinberg, S.E.; Hollister, S.J.; Das, S. Bone tissue engineering using polycaprolactone scaffolds fabricated via selective laser sintering. Biomaterials 2005, 26, 4817-4827, doi:10.1016/j.biomaterials.2004.11.057.
  10. Ba Linh, N.T.; Min, Y.K.; Lee, B.T. Hybrid hydroxyapatite nanoparticles-loaded PCL/GE blend fibers for bone tissue engineering. J Biomater Sci Polym Ed 2013, 24, 520-538, doi:10.1080/09205063.2012.697696.
  11. Ufere, S.K.J.; Sultana, N. Contact angle, conductivity and mechanical properties of polycaprolactone/hydroxyapatite/polypyrrole scaffolds using freeze-drying technique. ARPN J. Eng. Appl. Sci 2016, 11, 13686.
  12. Yang, X.; Yang, F.; Walboomers, X.F.; Bian, Z.; Fan, M.; Jansen, J.A. The performance of dental pulp stem cells on nanofibrous PCL/gelatin/nHA scaffolds. J Biomed Mater Res A 2010, 93, 247-257, doi:10.1002/jbm.a.32535.
  13. Aghazadeh, M.; Samiei, M.; Hokmabad, V.R.; Alizadeh, E.; Jabbari, N.; Seifalian, A.; Salehi, R. The Effect of Melanocyte Stimulating Hormone and Hydroxyapatite on Osteogenesis in Pulp Stem Cells of Human Teeth Transferred into Polyester Scaffolds. Fibers and Polymers 2018, 19, 2245-2253, doi:10.1007/s12221-018-8309-6.”

Round 2

Reviewer 4 Report

The authors have put their best efforts into improving the quality of the manuscript. All the comments are well addressed. The revised version of the manuscript can be accepted for publication.